# IPSeg: Image Posterior Mitigates Semantic Drift in Class-Incremental Segmentation

## Abstract

Class incremental learning aims to enable models to learn from sequential, non-stationary data streams across different tasks without catastrophic forgetting. In class incremental semantic segmentation (CISS), the semantic content of the background class changes across incremental phases, which is known as **semantic drift**. Our research identifies two severe issues within semantic drift: separate optimization and noisy semantics, which significantly degrade CISS performance. Based on this insight, we propose a simple yet effective method, **I**mage **P**osterior and Semantics Decoupling for **Seg**mentation (IPSeg), designed to address these challenges through two specific mechanisms. First, IPSeg leverages image posterior probabilities as guidance to resolve the separate optimization issue. Second, IPSeg utilizes semantics decoupling to effectively handle noisy semantics and tailor the learning strategies for different types of knowledge. Experiment results on the Pascal VOC 2012 and ADE20K datasets demonstrate superior performance compared to previous state-of-the-art approaches, particularly in more realistic and challenging long-term scenarios. Furthermore, IPSeg exhibits excellent properties in terms of both learning plasticity and memory stability.

## 1 Introduction

Deep learning methods have achieved significant success in vision (Qu et al., 2021) and language (Ke & Liu, 2022) tasks with fixed or stationary data distributions. However, real-world scenarios are characterized by dynamic and non-stationary data distributions, posing the challenge of *catastrophic forgetting* (McCloskey & Cohen, 1989; McClelland et al., 1995). Incremental learning, a.k.a. continual learning or lifelong learning (Silver et al., 2013), has been proposed to enable models to adapt to new data distributions without forgetting previous knowledge (Kudithipudi et al., 2022). Within this domain, Class Incremental Learning (CIL) methods (Serra et al., 2018; Li & Hoiem, 2017; Rebuffi et al., 2017; Mai et al., 2022; Wang et al., 2024a) have shown great potential in learning new classes from incoming data, particularly for classification tasks (De Lange et al., 2021).

Class Incremental Semantic Segmentation (CISS) extends the principles of CIL to pixel-wise tasks. In addition to catastrophic forgetting in CIL, CISS encounters an even more critical challenge: *semantic drift* (Yuan & Zhao, 2023) or *background shift* (Cermelli et al., 2020), which describes the incremental change in the semantic meaning of pixel labels. Several studies (Douillard et al., 2021; Cha et al., 2021; Zhang et al., 2022b; 2023) attribute *semantic drift* to the dynamic semantic content of the background across incremental stages. Subsequent works (Cermelli et al., 2020; Douillard et al., 2021) early pioneer this investigation using knowledge distillation and pseudo-labeling. More recent works (Cha et al., 2021; Zhang et al., 2022b; 2023) further use saliency maps and segment proposals to differentiate between the foreground and background regions. However, these works predominantly target the separation of *noisy semantics*, still leaving room for further optimization.

Furthermore, we delve into *semantic drift* challenge and identify an additional critical issue, *separate optimization*, as being of significant importance. *Separate optimization* refers to the CIL methods that update the task heads for each target class independently and sequentially. This leads to a scenario where task heads trained earlier may produce higher scores than those trained later for similar-looking categories. Figure 1(a) directly presents the impact of *separate optimization*, where the SSUL-M model mistakenly classifies a horse as a "cow" class with a higher logit score after learning "horse"

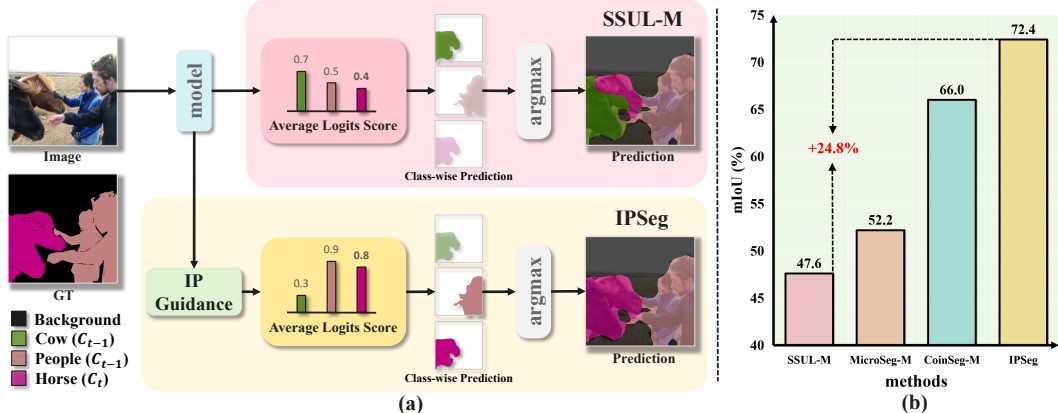

Figure 1: (a) Affected by *noisy semantics* and *separate optimization*, the previous method SSUL-M misclassifies a "horse" as a "cow" with higher logit scores when learning "horse" after "cow". While our IPSeg leverages image posterior (IP) guidance to produce accurate predictions on these two similar-look classes. In class-wise prediction visualizations, we use deeper colors to indicate higher prediction scores. (b) The quantitative performance comparison with state-of-the-art methods under the long-term incremental challenge (VOC 2-2).

following "cow". Under the combined impacts of *separate optimization* and *noisy semantics*, the previous efforts are still short of effectively addressing the *semantic drift* challenge.

Motivated by our observations and analyses, we introduce **I**mage **P**osterior and Semantics Decoupling for Class-Incremental Semantic **Seg**mentation (IPSeg) to address the aforementioned challenges. Specifically, we propose **image posterior guidance** to mitigate *separate optimization* by rectifying the error pixel-wise predictions using image-wise predictions. As illustrated in Figure 1(a), IPSeg is capable of correctly predicting "horse" with the assistance of image posterior guidance. Moreover, we propose **permanent-temporary semantics decoupling** to decouple the *noisy semantics* into two groups, one characterized by simple, stable, and static semantics, and the other by complex, dynamic, and temporary semantics. To accommodate these distinct semantic groups, we design separate permanent and temporary branches with varied life cycles to learn the associated concepts.

Extensive experimental results on two popular benchmarks, Pascal VOC 2012 and ADE20K, demonstrate the effectiveness and competitiveness of IPSeg. Our method consistently outperforms other methods across various incremental scenarios, particularly in long-term challenges with performance gains of **24.8**% in VOC 2-2 task as illustrated in Figure 1(b). Experiment results further reveal that IPSeg has good properties of both learning plasticity and memory stability.

## 2 RELATED WORK

**Class Incremental Learning (CIL)**  Class-incremental learning is a method that continuously acquires knowledge in the order of classes, aiming to address catastrophic forgetting (McCloskey & Cohen, 1989) while continually learning new classes. Existing work (Wang et al., 2024b) broadly categorizes these approaches into three main types: **Replay-based methods** involve storing data or features of old classes or generating data that includes old classes to mitigate catastrophic forgetting. This class can be further divided into Experience Replay (Rebuffi et al., 2017; Bang et al., 2021), Generative Replay (Liu et al., 2020; Shin et al., 2017), and Feature Replay (Belouadah & Popescu, 2019). **Regularization-based methods** focus on designing loss functions that incorporate second-order penalties based on the contribution of parameters to different tasks (Kirkpatrick et al., 2017; Jung et al., 2020). They also rely on knowledge distillation, typically using the model from a previous phase as a teacher to constrain the current phase model (Li & Hoiem, 2017; Rebuffi et al., 2017; Douillard et al., 2020; Buzzega et al., 2020). **Architecture-based methods** dynamically adjust model parameters based on new data, including assigning specific parameters for different data (Gurbuz & Dovrolis, 2022; Serra et al., 2018) and breaking down model parameters into task-specific or shared parts (Douillard et al., 2022).

**Class Incremental Semantic Segmentation (CISS)** CISS is similar to class-incremental learning (CIL) but extends the task to pixel-level predictions (Phan et al., 2022; Camuffo & Milani, 2023; Zhang et al., 2022a; Xiao et al., 2023). MiB (Cermelli et al., 2020) first introduces the concept of semantic shift unique to CISS, employing distillation strategies to mitigate this issue. PLOP (Douillard et al., 2021) utilizes pseudo-labeling techniques for incremental segmentation to address background shift, while SSUL (Cha et al., 2021) further incorporates salient information, introducing the concept of "unknown classes" into each learning phase and using a memory pool to store old data to prevent catastrophic forgetting. RECALL (Maracani et al., 2021) and DiffusePast (Chen et al., 2023) extend traditional replay methods by incorporating synthetic samples of previous classes generated using Diffusion (Ho et al., 2020) or GAN (Goodfellow et al., 2020) models. MicroSeg (Zhang et al., 2022b) employs a proposal generator to simulate unseen classes. CoinSeg (Zhang et al., 2023) highlights differences within and between classes, designing a contrastive loss to adjust the feature distribution of classes. PFCSS (Lin et al., 2023) emphasizes the preemptive learning of future knowledge to enhance the model's discrimination ability between new and old classes.

## 3 METHOD

In this section, we begin by presenting the necessary notation and definition of the problem, followed by our analysis of *semantic drift* in section 3.1. Next, we introduce our proposed method, IPSeg, with detailed designs including image posterior and semantics decoupling in section 3.3 and section 3.4.

### 3.1 PRELIMINARY

**Notation and problem formulation** Following previous works (Cha et al., 2021; Zhang et al., 2022b; 2023), in CISS, a model needs to learn the target classes $\mathcal{C}_{1:T}$ from a series of incremental tasks as $t = 1, 2, 3, ..., T$. For task $t$, the model learns from a unique training dataset $\mathcal{D}_t$ which consists of training data and ground truth pairs $\mathcal{D}_t = \{(x_i^t, y_i^t)\}_{i=1}^{|\mathcal{D}_t|}$. Here $i$ denotes the sample index, $t$ for the task index, and $|\mathcal{D}_t|$ for the training dataset scale. $x_{i,j}^t$ and $y_{i,j}^t$ denote the $j$-th pixels and the corresponding annotation in the image $x_i^t$. In each incremental phase $t$, the model can only access the class set $\mathcal{C}_t \cup c_b$ where $\mathcal{C}_t$ denotes the class set of current task $t$ and $c_b$ for background class.

To prevent catastrophic forgetting, architecture-based methods allocate and optimize distinct sets of parameters for each class, instead of directly updating the whole model. The whole model $f_t$ is composed of a frozen backbone $h_\theta$, followed by a series of learnable task-specific heads $\phi_{1:t}$, with one task head corresponding to each task. In task $t$, only the newly added task head $\phi_t$ is needed to be optimized. In inference, the prediction for the $j$-th pixel in image $x_i$ can be obtained by:

$$\hat{y}_{i,j} = f_t(x_{i,j}) = \underset{c \in \mathcal{C}_{1:T}}{\arg\max} \, \phi_{1:T}^c(h_\theta(x_{i,j})). \tag{1}$$

Where $\phi_{1:T}^c(\cdot)$ denotes the $C$-dimension outputs. Additionally, we introduce the image-level labels $\mathcal{Y}_i$ of the image $x_i$, a memory buffer $\mathcal{M}$, and an extra image classification head $\psi$ to support our implementation. A comprehensive explanation of symbols can be found in the appendix.

**Semantic Drift** Previous works (Kirkpatrick et al., 2017) mainly attribute the *semantic drift* to *noisy semantics* within the background class $c_b$. They attempt to mitigate this challenge by decoupling the background class $c_b$ into subclasses $c_b'$ and $c_u$, where $c_b'$ denotes the "pure" background and $c_u$ denotes the unknown class. The most advanced methods (Zhang et al., 2022b; Cha et al., 2021) further decouple the unknown classes $c_u$ into past seen classes $\mathcal{C}_{1:t-1}$ and dummy unknown class $c_u'$ using pseudo labeling. However, *semantic drift* remains unresolved as the decoupled classes are still changing across incremental phases and models are always hard to learn these chaotic classes.

Additionally, another challenge caused by *separate optimization* of incremental learning exacerbates *semantic drift* but attracts few attention. Previous work (Kim et al., 2024) points out that freezing parameters from the old stage can preserve the model's prior knowledge, leading to error propagation and confusion between similar classes. While in CISS, the task head $\phi_t$ is exclusively trained by supervision from the current classes and will be frozen to resist catastrophic forgetting in the following incremental phases. In the following task $t_1, t_1 > t$, $\phi_t$ may produce high scores on objects from other appearance-similar classes, without any penalty and optimization. Meanwhile, the new task head $\phi_{t_1}$ just predicts moderate scores which might be slightly lower than error predictions from $\phi_t$.

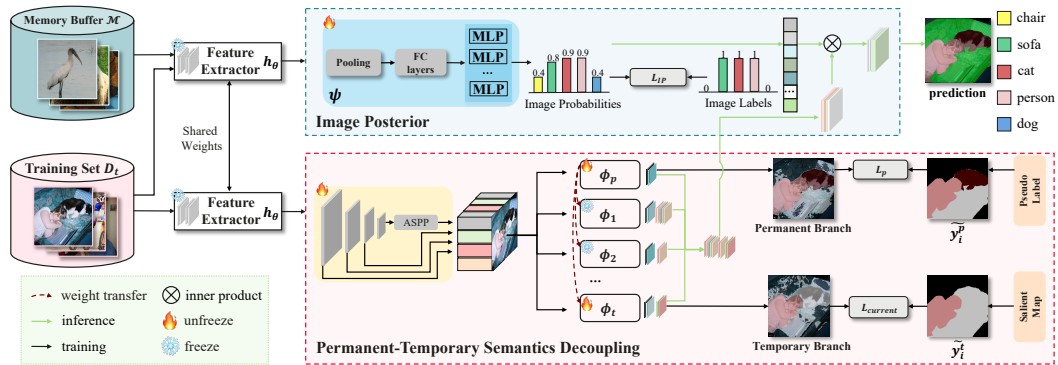

Figure 2: Overall architecture of our proposed IPSeg, mainly composed of image posterior and permanent-temporary semantics decoupling two parts. In the latter part, $\phi_p$ denotes the permanent learning branch and $\phi_1, \phi_2, ..., \phi_t$ for temporary ones. The black solid lines are used to indicate the data flow in training and the green ones are for inference.

In this way, it is common that earlier incremental task heads may have larger output scales than the later heads, especially in similar classes. This *separate optimization* manner ultimately causes the incremental models to misclassify some categories and makes *semantic drift* more difficult to address. Some cases are provided to help better understand this challenge in the appendix.

## 3.2 OVERVIEW

As illustrated in Figure 2, we propose **I**mage **P**osterior and Semantics Decoupling for Class-incremental Semantic **Seg**mentation (IPSeg) to mitigate *semantic drift* through two main strategies: image posterior guidance and permanent-temporary semantics decoupling. In section 3.3, we describe how the IPSeg model uses image posterior guidance to overcome *separate optimization*. To address *noisy semantics*, IPSeg employs two branches to decouple the learning of noisy semantics. Detailed explanations of this approach are provided in section 3.4.

## 3.3 IMAGE POSTERIOR GUIDANCE

As previously discussed, the *separate optimization* leads to inconsistent output scales across different incremental task heads and error predictions. We propose leveraging the image-level posterior as the global guidance to correct the probability distributions of different task heads. The rationale for using the image posterior probabilities is based on the following fact:

**Fact**: *For any image, if its image-level class domain is $\mathcal{C}_I$ and its pixel-level class domain is $\mathcal{C}_P$, the class domains $\mathcal{C}_I$ and $\mathcal{C}_P$ are the same, i.e., $\mathcal{C}_I = \mathcal{C}_P$.*

Inspired by this fact, we propose to use an extra image posterior branch $\psi$ to predict image classification labels and train it in an incremental learning manner. As illustrated in Figure 2, $\psi$ is composed of Pooling, Fully connected (FC) layers, and Multi-Layer Perceptrons (MLPs) with the input dimension of 4096 and the output dimension of $|\mathcal{C}_{1:T}|$, where the FC layers serve as shared intermediate feature processors, and the MLPs serve as incremental classification heads for incremental classes.

In task $t$ ($t > 1$), the model is only able to access data from both the memory buffer $\mathcal{M}$ and the current task training data $\mathcal{D}_t$. The mixed data samples $x_i^{m,t}$ from these two sources are processed by the network backbone $h_\theta$ into the image feature $h_\theta(x_i^t)$, and further processed by image posterior branch $\psi$ into the image classification prediction $\hat{\mathcal{Y}}_i^t$. The objective function for training $\psi$ is:

$$\mathcal{L}_{\text{IP}} = \mathcal{L}_{\text{BCE}}(\hat{\mathcal{Y}}_i^t, \tilde{\mathcal{Y}}_i^t) = \mathcal{L}_{\text{BCE}}(\psi(h_\theta(x_i^{m,t})), \tilde{\mathcal{Y}}_i^t), \ \ \tilde{\mathcal{Y}}_i^t = \mathcal{Y}_i^t \cup \tilde{\mathcal{Y}}_{\phi_{1:t-1}(h_\theta(x_i^{m,t}))}. \tag{2}$$

Where image classification label $\tilde{\mathcal{Y}}_i^t$ consists of two parts, the ground truth label $\mathcal{Y}_i^t$ on current task class set $\mathcal{C}_t$ and pseudo label $\tilde{\mathcal{Y}}_{\phi_{1:t-1}(h_\theta(x_i^{m,t}))}$ on past seen classes $\mathcal{C}_{1:t-1}$. Instead of relying solely on the label $\mathcal{Y}_i^t$, we use the image-level pseudo labels from previous task heads prediction to provide informative signals of previous incremental tasks, mitigating biases in the image posterior branch $\psi$.

During testing, the image posterior branch predicts posterior probabilities on all classes $\mathcal{C}_{1:T}$. For a test image $x_i$, the final pixel-wise prediction scores are computed by element-wise multiplication between the image posterior probabilities from $\psi$ and the pixel-wise probabilities from $\phi_{0:T}$:

$$p_i = \underbrace{\texttt{Concat}(\ \alpha_{\text{BC}},\ \sigma(\ \psi(h_\theta(x_i))\ )\ )}_{\text{Image Posterior Probability}} \cdot \sigma(\ \underbrace{\phi_{0:T}(h_\theta(x_i))}_{\text{Pixel-wise Probability}}\ ). \tag{3}$$

where $\sigma(\cdot)$ denotes the Sigmoid function. The hyperparameter $\alpha_{\text{BC}}$ is used to compensate for the lack of background posterior probability, with the default value $\alpha_{\text{BC}} = 0.9$. The result $p_i$ is the rectified pixel-wise prediction with a shape of $[C, HW]$, and $p_{i,j}^c$ is prediction of the $j$-th pixel on class $c$. The final prediction of the $j$-th pixel by IPSeg can be written as:

$$\hat{y}_{i,j} = \underset{c \in \mathcal{C}_{1:t}}{\arg\max}\, p_{i,j}^c. \tag{4}$$

## 3.4 PERMANENT-TEMPORARY SEMANTICS DECOUPLING

To further address *semantic drift* caused by the coupled learning of complex and noisy dummy labels $c_b$ and $c_u$ along with incomplete yet accurate label $\mathcal{C}_t$, we propose a decoupling strategy that segregates the learning process for different semantics. Here is our empirical observation:

**Observation**: *Given an image in incremental task t, the semantic contents of it include four parts: the region of past classes $\mathcal{C}_{1:t-1}$, target classes $\mathcal{C}_t$, unknown foreground $c_u'$ and pure background $c_b'$.*

Based on this observation, we first introduce dummy label $c_f = \mathcal{C}_{1:t-1} \cup c_u'$ to represent the foreground regions that encompass both past seen classes and unknown classes, which are not the primary targets in the current task. Subsequently, we decouple the regions of a training image into two sets: $\mathcal{C}_t \cup c_f$ and $c_b' \cup c_u'$. The former set $\mathcal{C}_t \cup c_f$ are current target classes and other foreground objects, which are temporary concepts varying across different phases. While $c_b' \cup c_u'$ are dummy labels representing "pure" background and unknown objects, which are permanent concepts existing throughout all incremental phases. For instance, "cat" and "horse" are target classes in the current task but change into the region of past seen foreground classes in subsequent tasks, while the concepts of background and unknown objects remain consistent.

Following our decoupling strategy, we can reassign the labels of image $x_i$ as:

$$\tilde{y}_i^p = \begin{cases} c_i, & \text{if } y_i^t \in \mathcal{C}_t \vee ((y_i^t = c_b) \wedge (f_{t-1}(x_i) \in \mathcal{C}_{1:t-1})) \\ c_u', & \text{if } (y_i^t = c_b) \wedge (f_{t-1}(x_i) \notin \mathcal{C}_{1:t-1}) \wedge (S(x_i) = 1) \\ c_b, & \text{else,} \end{cases} \quad \tilde{y}_i^t = \begin{cases} y_i^t, & \text{if } y_i^t \in \mathcal{C}_t \\ c_f, & \text{if } (y_i^t = c_b) \wedge (S(x_i) = 1) \\ c_b', & \text{else,} \end{cases} \tag{5}$$

where $f_{t-1}(\cdot)$ is the model of task $t-1$ and $S(\cdot)$ is the salient object detector as used in SSUL (Cha et al., 2021). $\tilde{y}_i^p$ is the label used to train the permanent branch $\phi_p$. $\tilde{y}_i^t$ is the label used to train the incremental learning heads $\phi_t$ for the current task $t$. $c_i$ is the ignored region not included in the loss calculation. The visualization of semantics decoupling is provided in the appendix.

The learning of these two sets of semantics is also decoupled. In addition to the existing incremental heads $\phi_{1:T}$, we introduce a permanent branch $\phi_p$ to learn the permanent dummy classes $c_b'$ and $c_u'$. $\phi_p$ has the same network architecture as the incremental head $\phi_t$. They are composed of three 3x3 convolution layers and several upsampling layers. The existing incremental head $\phi_t$ serves as the temporary branch to learn the semantics $\mathcal{C}_t \cup c_f$ existing in the current incremental phase. It's worth noting that $\phi_p$ and $\phi_t$ have different learning life cycles. The permanent branch $\phi_p$ is trained and optimized across all incremental phases to distinguish foreground and background. While temporary branch $\phi_t$ is temporarily trained in task phase $t$ to recognize the current foreground classes $\mathcal{C}_t$. The objective functions for these two branches is defined as:

$$\mathcal{L}_p = \mathcal{L}_{\text{BCE}}(\ \phi_p(h_\theta(x_i^t)), \tilde{y}_i^p\ ), \quad \mathcal{L}_{\text{current}} = \mathcal{L}_{\text{BCE}}(\ \phi_t(h_\theta(x_i^t)), \tilde{y}_i^t\ ). \tag{6}$$

Finally, the total optimization objective function of IPSeg is:

$$\mathcal{L}_{total} = \mathcal{L}_{\text{IP}} + \lambda_1 \mathcal{L}_{\text{current}} + \lambda_2 \mathcal{L}_p, \tag{7}$$

where $\lambda_1$ and $\lambda_2$ are trade-off hyperparameters to balance different training objective functions.

During inference, as shown by the greens line in Figure 2, the permenant branch $\phi_p$ outputs prediction for the background $c_b'$, and the temporary branch $\phi_i$ ($i = 1, 2, ..., t$) outputs predictions for the target classes $\mathcal{C}_t$. And the pixel-level prediction $\phi_{0:T}(h_\theta(x_i^t))$ can be writen as:

$$\phi_{0:T}(h_\theta(x_i^t)) = \texttt{Concat}(\ \phi_p(h_\theta(x_i^t)),\ \phi_{1:T}(h_\theta(x_i^t))). \tag{8}$$

Where $\phi_p(h_\theta(x_i^t))$ and $\phi_{1:T}(h_\theta(x_i^t))$ represent background prediction from permenant branch and all foreground prediction from temporary branch. The pixel-level prediction is then producted by image posterior probability to form the final predition maps as Eq 3 and Eq 4.

Furthermore, to mitigate the issue of inaccurate predictions on other foreground classes $c_f$ within each task head $\phi_t$ during inference, we introduce a Noise Filtering trick, filtering out prediction errors associated with $c_f$. The prediction for the $j$-th pixel $\hat{y}_{i,j}$ is processed as:

$$\hat{y}_{i,j} = \begin{cases} \alpha_{\text{NF}} \cdot \hat{y}_{i,j} & \text{if } max(\, p_{i,j}^f, \, p_{i,j}^c \,) = p_{i,j}^f \\ \hat{y}_{i,j} & \text{if } max(\, p_{i,j}^f, \, p_{i,j}^c \,) = p_{i,j}^c \end{cases} \tag{9}$$

Where $\alpha_{\text{NF}}$ is noise filtering term with the default value $\alpha_{\text{NF}} = 0.4$. And $p_{i,j}^f$ and $p_{i,j}^c$ are the $j$-th pixel logit outputs on the foreground and target class respectively.

### 3.5 Improving Memory Buffer

The memory buffer $\mathcal{M}$ plays a crucial role in our implementation and we implement the memory buffer based on unbiased learning and storage efficiency. IPSeg employs a class-balanced sampling strategy, ensuring the image posterior branch can adequately access samples from all classes. Specifically, given the memory size $|\mathcal{M}|$ and the number of already seen classes $|\mathcal{C}_{1:t}|$, the sampling strategy ensures there are at least $|\mathcal{M}| \, // \, |\mathcal{C}_{1:t}|$ samples for each class. IPSeg also optimizes the storage cost of $\mathcal{M}$ by only storing image-level labels and object salient masks for samples. Image-level labels are required for the image posterior branch for unbiased classification. While the salient masks split images into background and foreground objects, labeled with 0 and 1 respectively. This simplification mechanism requires less storage cost compared to previous methods that store the whole pixel-wise annotations on all classes. More details can be found in the appendix.

## 4 Experiments

### 4.1 Experimental Setups

**Dataset** Following previous works (Zhang et al., 2023; Cha et al., 2021), we evaluate our method using the Pascal VOC 2012 (Everingham et al., 2010) and ADE20K (Zhou et al., 2017) datasets. Pascal VOC 2012 includes 20 foreground classes and one background class, with 10,582 training images and 1,449 validation images. ADE20K, a larger-scale dataset, comprises 150 classes of stuff and objects, with 20,210 training images and 2,000 validation images.

**Protocols** We primarily use the *overlap* setting to evaluate our method. This setting is more challenging and realistic than the *disjoint* setting (Qiu et al., 2023), as the images may contain both seen and unseen classes across different incremental steps. We evaluate IPSeg under several incremental scenarios, denoted as $M$-$N$, where $M$ is the number of classes learned initially, and $N$ is the number of classes learned in each incremental step. For example, VOC 15-1 (6 steps) means learning 15 classes initially and one new class in each subsequent step until all 20 classes are learned. We use the mean Intersection over Union (mIoU) as the evaluation metric.

**Implementation details** Following previous works (Lin et al., 2022; Baek et al., 2022; Xiao et al., 2023), IPSeg utilizes DeepLab V3 (Chen et al., 2017) as the segmentation model with ResNet-101 (He et al., 2016) and Swin Transformer-base (Swin-B) (Liu et al., 2021) pre-trained on ImageNet-1K (Deng et al., 2009) as the backbones. The training batch size is 16 for Pascal VOC 2012 and 8 for ADE20K. IPSeg uses the SGD optimizer with a momentum of 0.9 and a weight decay of $1e$-4. The learning rates for both datasets are set to 0.01, with learning rate policies of poly for Pascal VOC 2012 and warm poly for ADE20K. All experiments are conducted with 2 NVIDIA GeForce RTX 3090 GPUs. For a fair comparison, the memory size is set as the same as SSUL (Cha et al., 2021) that $|\mathcal{M}| = 100$ for Pascal VOC 2012 and $|\mathcal{M}| = 300$ for ADE20K. Pseudo-label (Zhang et al., 2023) and saliency information (Hou et al., 2017) are adopted as previous methods (Cha et al., 2021; Zhang et al., 2022b). To avoid information leaking, the ground truth of the training data in the image posterior branch only consists of annotations and pseudo-labels from the corresponding steps.

Table 1: Comparison with state-of-the-art methods on Pascal VOC 2012 dataset across 4 typical incremental segmentation scenarios. "-" denotes the results not provided in the original paper. [†] denotes the result is reproduced using the official code with Swin-B backbone. "IPSeg w/o M" denotes the data-free version of IPSeg, which is trained without the memory buffer.

| | Method | Backbone | VOC 15-5 (2 steps) | | | VOC 15-1 (6 steps) | | | VOC 10-1 (11 steps) | | | VOC 2-2 (10 steps) | | |
|---|---|---|---|---|---|---|---|---|---|---|---|---|---|---|
| | | | 0-15 | 16-20 | all | 0-15 | 16-20 | all | 0-10 | 11-20 | all | 0-2 | 3-20 | all |
| Data-free | LwF-MC (Rebuffi et al., 2017) | Resnet-101 | 58.1 | 35.0 | 52.3 | 6.4 | 8.4 | 6.9 | 4.7 | 5.9 | 5.0 | 3.5 | 4.7 | 4.5 |
| | ILT (Michieli & Zanuttigh, 2019) | Resnet-101 | 67.1 | 39.2 | 60.5 | 8.8 | 8.0 | 8.6 | 7.2 | 3.7 | 5.5 | 5.8 | 5.0 | 5.1 |
| | MiB (Cermelli et al., 2020) | Resnet-101 | 71.8 | 43.3 | 64.7 | 46.2 | 12.9 | 37.9 | 12.3 | 13.1 | 12.7 | 41.1 | 23.4 | 25.9 |
| | SSUL (Cha et al., 2021) | Resnet-101 | 77.8 | 50.1 | 71.2 | 77.3 | 36.6 | 67.6 | 71.3 | 46.0 | 59.3 | 62.4 | 42.5 | 45.3 |
| | RCIL (Zhang et al., 2022a) | Resnet-101 | **78.8** | 52.0 | 72.4 | 70.6 | 23.7 | 59.4 | 55.4 | 15.1 | 34.3 | 28.3 | 19.0 | 19.4 |
| | IDEC (Zhao et al., 2023) | Resnet-101 | 78.0 | 51.8 | 71.8 | 77.0 | 36.5 | 67.3 | 70.7 | 46.3 | 59.1 | - | - | - |
| | PLOP+LGKD (Yang et al., 2023) | Resnet-101 | 75.2 | 54.8 | 71.1 | 69.3 | 30.9 | 61.1 | - | - | - | - | - | - |
| | PLOP+NeST (Xie et al., 2024) | Resnet-101 | 77.6 | **55.8** | 72.4 | 72.2 | 33.7 | 63.1 | 54.2 | 17.8 | 36.9 | - | - | - |
| | IPSeg w/o M (ours) | Resnet-101 | 78.5 | 55.2 | **72.9** | **77.4** | **41.9** | **68.9** | **74.9** | **52.9** | **64.4** | **64.7** | **51.5** | **53.4** |
| | SSUL (Cha et al., 2021) | Swin-B | 79.7 | 55.3 | 73.9 | 78.1 | 33.4 | 67.5 | 74.3 | 51.0 | 63.2 | 60.3 | 40.6 | 44.0 |
| | MicroSeg (Zhang et al., 2022b) | Swin-B | **81.9** | 54.0 | 75.2 | 80.5 | 40.8 | 71.0 | 73.5 | 53.0 | 63.8 | 64.8 | 43.4 | 46.5 |
| | PLOP+NeST (Xie et al., 2024) | Swin-B | 80.5 | **70.8** | **78.2** | 76.8 | **57.2** | 72.2 | 64.3 | 28.3 | 47.3 | - | - | - |
| | IPSeg w/o M (ours) | Swin-B | 81.4 | 62.4 | 76.9 | **82.4** | 52.9 | **75.4** | **80.0** | **61.2** | **71.0** | **72.1** | **64.5** | **65.5** |
| Replay | Joint | Resnet-101 | 80.5 | 73.0 | 78.2 | 80.5 | 73.0 | 78.2 | 79.1 | 77.1 | 78.2 | 73.9 | 78.9 | 78.2 |
| | SDR (Michieli & Zanuttigh, 2021) | Resnet-101 | 75.4 | 52.6 | 69.9 | 44.7 | 21.8 | 39.2 | 32.4 | 17.1 | 25.1 | 13.0 | 5.1 | 6.2 |
| | PLOP-M (Douillard et al., 2021) | Resnet-101 | 78.5 | 65.6 | 75.4 | 71.1 | 52.6 | 66.7 | 57.9 | 51.6 | 54.9 | - | - | - |
| | SSUL-M (Cha et al., 2021) | Resnet-101 | 79.5 | 52.9 | 73.2 | 78.9 | 43.9 | 70.6 | 74.8 | 48.9 | 65.5 | 58.8 | 45.8 | 47.6 |
| | MicroSeg-M (Zhang et al., 2022b) | Resnet-101 | **82.0** | 59.2 | 76.6 | **81.3** | 52.5 | 74.4 | **77.2** | 57.2 | 67.7 | 60.0 | 50.9 | 52.2 |
| | PFCSS-M (Lin et al., 2023) | Resnet-101 | 79.9 | 70.2 | 77.1 | 77.1 | **60.4** | 73.1 | 69.5 | 63.2 | 66.5 | - | - | - |
| | IPSeg (ours) | Resnet-101 | 79.5 | **71.0** | **77.5** | 79.6 | 58.9 | **74.7** | 75.9 | **66.4** | **71.4** | **62.4** | **61.0** | **61.2** |
| | Joint | Swin-B | 83.8 | 79.3 | 82.7 | 83.8 | 79.3 | 82.7 | 82.4 | 83.0 | 82.7 | 75.8 | 83.9 | 82.7 |
| | SSUL-M† (Cha et al., 2021) | Swin-B | 79.3 | 55.1 | 73.5 | 78.8 | 49.7 | 71.9 | 75.3 | 54.1 | 65.2 | 61.1 | 47.5 | 49.4 |
| | MicroSeg-M† (Zhang et al., 2022b) | Swin-B | 82.9 | 60.1 | 77.5 | 82.0 | 47.3 | 73.3 | 78.9 | 59.2 | 70.1 | 62.7 | 51.4 | 53.0 |
| | CoinSeg-M (Zhang et al., 2023) | Swin-B | **84.1** | 69.9 | 80.8 | **84.1** | 65.5 | 79.6 | **81.3** | 64.4 | 73.7 | 68.4 | 65.6 | 66.0 |
| | IPSeg (ours) | Swin-B | 83.3 | **73.3** | **80.9** | 83.5 | **75.1** | **81.5** | 80.3 | **76.7** | **78.6** | **73.1** | **72.3** | **72.4** |

**Baselines** We compare IPSeg with various data-free and replay-based CISS methods, including LwF-MC (Rebuffi et al., 2017), ILT (Michieli & Zanuttigh, 2019), MiB (Cermelli et al., 2020), SDR (Michieli & Zanuttigh, 2021), and PLOP (Douillard et al., 2021), as well as state-of-the-art methods such as RCIL (Zhang et al., 2022a), IDEC (Zhao et al., 2023), SSUL (Cha et al., 2021), MicroSeg (Zhang et al., 2022b), PFCSS (Lin et al., 2023), CoinSeg (Zhang et al., 2023) and NeST (Xie et al., 2024). Among these, PFCSS, CoinSeg, and NeST are the current state-of-the-art. For a fair comparison, we reproduce some works using their official code with the Swin-B backbone. Additionally, we provide the results of **Joint** as a theoretical upper bound for incremental tasks. We report incremental results in three parts: initial classes, new classes, and overall classes.

## 4.2 MAIN RESULTS

**Results on Pascal VOC 2012** We evaluate IPSeg in various incremental scenarios on Pascal VOC 2012, including standard incremental scenarios (15-5 and 15-1) and long-term incremental scenarios (10-1 and 2-2). As shown in Table 1, IPSeg achieves the best results across all incremental scenarios on Pascal VOC 2012 with both ResNet-101 and Swin-B backbones. Notably, in the long-term incremental scenarios 10-1 and 2-2, IPSeg achieves performance gains of **4.9**% and **6.4**% over the second-best method, CoinSeg-M, with the same Swin-B backbone. Meanwhile, the data-free version of IPSeg (denotes as "IPSeg w/o M" ) also demonstrates competitive performance.

The superior and robust performance of IPSeg is mainly attributed to the reliable role of guidance provided by the image posterior branch. The image posterior design effectively helps IPSeg avoid catastrophic forgetting and achieve excellent performance on new classes, which often suffer from semantic drift due to separate optimization in new steps. Additionally, the semantics decoupling design enables IPSeg to better learn foreground classes within each incremental step. This design brings the improvement of **12.3**% and **6.7**% over CoinSeg-M on new classes in the 10-1 and 2-2 scenarios, respectively.

Furthermore, as illustrated in Figure 3(a), IPSeg experiences less performance degradation compared to previous state-of-the-art methods as the number of incremental steps increases, which indicates that IPSeg has stronger resistance to catastrophic forgetting. This conclusion is further supported by the data in Figure 3(b) and Figure 3(c), which show that IPSeg exhibits minimal performance declines as the incremental process continues. In contrast, other methods only maintain comparable performance during the initial incremental learning step but quickly degrade in subsequent steps

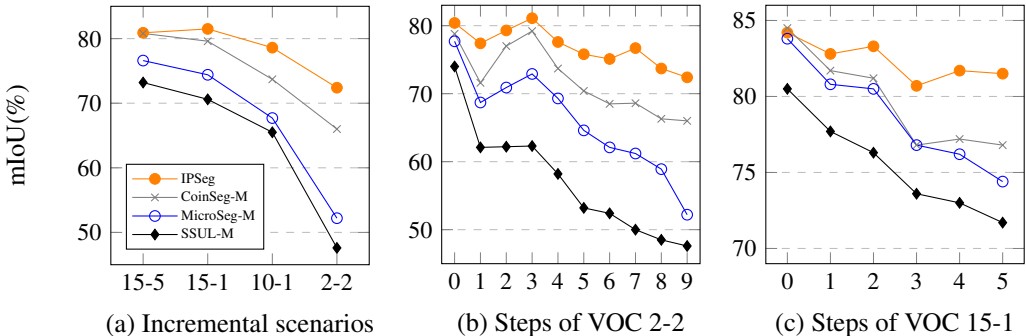

|  | (a) Incremental scenarios | (b) Steps of VOC 2-2 | (c) Steps of VOC 15-1 |

Figure 3: (a) The overall performance of different methods on Pascal VOC 2012 under 4 scenarios, (b) mIoU visualization on Pascal VOC 2012 2-2, (c) mIoU visualization on Pascal VOC 2012 15-1.

Table 2: Comparison with state-of-the-art methods on ADE20K dataset. $^{\dagger}$ denotes the result is reproduced using the official code with Swin-B backbone.

| | Method | Backbone | ADE 100-5 (11 steps) | | | ADE 100-10 (6 steps) | | | ADE 100-50 (2 steps) | | | ADE 50-50 (3 steps) | | |
|---|---|---|---|---|---|---|---|---|---|---|---|---|---|---|
| | | | 0-100 | 101-150 | all | 0-100 | 101-150 | all | 0-100 | 101-150 | all | 0-50 | 51-150 | all |
| Data-free | MiB (Cermelli et al., 2020) | Resnet-101 | 36.0 | 5.7 | 26.0 | 38.2 | 11.1 | 29.2 | 40.5 | 17.2 | 32.8 | 45.6 | 21.0 | 29.3 |
| | SSUL (Cha et al., 2021) | Resnet-101 | 39.9 | 17.4 | 32.5 | 40.2 | 18.8 | 33.1 | 41.3 | 18.0 | 33.6 | 48.4 | 20.2 | 29.6 |
| | RCIL (Zhang et al., 2022a) | Resnet-101 | 38.5 | 11.5 | 29.6 | 39.3 | 17.6 | 32.1 | 42.3 | 18.8 | 34.5 | 48.3 | 25.0 | 32.5 |
| | MicroSeg (Zhang et al., 2022b) | Resnet-101 | 40.4 | 20.5 | 33.8 | 41.5 | 21.6 | 34.9 | 40.2 | 18.8 | 33.1 | 48.6 | 24.8 | 32.9 |
| | IDEC (Zhao et al., 2023) | Resnet-101 | 39.2 | 14.6 | 31.0 | 40.3 | 17.6 | 32.7 | 42.0 | 18.2 | 34.1 | 47.4 | 26.0 | 33.1 |
| | AWT+MiB (Goswami et al., 2023) | Resnet-101 | 38.6 | 16.0 | 31.1 | 39.1 | 21.4 | 33.2 | 40.9 | 24.7 | 35.6 | 46.6 | 27.0 | 33.5 |
| | PLOP+LGKD (Yang et al., 2023) | Resnet-101 | - | - | - | 42.1 | 22.0 | 35.4 | 43.6 | 25.7 | 37.5 | 49.4 | 29.4 | 36.0 |
| | PLOP+NeST (Xie et al., 2024) | Resnet-101 | 39.3 | 17.4 | 32.0 | 40.9 | 22.0 | 34.7 | 42.2 | 24.3 | 36.3 | 48.7 | 27.7 | 34.8 |
| | IPSeg w/o M (ours) | Resnet-101 | 41.0 | 22.4 | 34.8 | 41.0 | 23.6 | 35.3 | 41.3 | 24.0 | 35.5 | 46.7 | 26.2 | 33.1 |
| | SSUL$^{\dagger}$ (Cha et al., 2021) | Swin-B | 41.3 | 16.0 | 32.9 | 40.7 | 19.0 | 33.5 | 41.9 | 20.1 | 34.6 | 49.5 | 21.3 | 30.7 |
| | CoinSeg (Zhang et al., 2023) | Swin-B | 43.1 | 24.1 | 36.8 | 42.1 | 24.5 | 36.2 | 41.6 | 26.7 | 36.6 | 49.0 | 28.9 | 35.6 |
| | PLOP+NeST (Xie et al., 2024) | Swin-B | 39.7 | 18.3 | 32.6 | 41.7 | 24.2 | 35.9 | 43.5 | 26.5 | 37.9 | 50.6 | 28.9 | 36.2 |
| | IPSeg w/o M (ours) | Swin-B | 43.1 | 26.2 | 37.6 | 42.5 | 27.8 | 37.6 | 43.2 | 29.0 | 38.4 | 49.3 | 33.0 | 38.5 |
| Replay | Joint | Resnet-101 | 43.5 | 29.4 | 38.3 | 43.5 | 29.4 | 38.8 | 43.5 | 29.4 | 38.8 | 50.3 | 32.7 | 38.8 |
| | SSUL-M (Cha et al., 2021) | Resnet-101 | 42.9 | 17.8 | 34.6 | 42.9 | 17.7 | 34.5 | 42.8 | 17.5 | 34.4 | 49.1 | 21.0 | 29.8 |
| | TIKP (Yu et al., 2024) | Resnet-101 | 37.5 | 17.6 | 30.9 | 41.0 | 19.6 | 33.8 | 42.2 | 20.2 | 34.9 | 48.8 | 25.9 | 33.6 |
| | IPSeg (ours) | Resnet-101 | 42.4 | 22.7 | 35.9 | 42.1 | 22.3 | 35.6 | 41.7 | 25.2 | 36.3 | 47.3 | 26.7 | 33.6 |
| | Joint | Swin-B | 47.2 | 31.9 | 42.1 | 47.2 | 31.9 | 42.1 | 47.2 | 31.9 | 42.1 | 54.6 | 35.5 | 42.1 |
| | SSUL-M$^{\dagger}$ (Cha et al., 2021) | Swin-B | 41.6 | 20.1 | 34.5 | 41.6 | 19.9 | 34.4 | 41.5 | 48.0 | 33.7 | 47.6 | 18.8 | 28.5 |
| | CoinSeg-M$^{\dagger}$ (Zhang et al., 2023) | Swin-B | 42.8 | 24.8 | 36.8 | 39.6 | 24.8 | 34.7 | 38.7 | 23.7 | 33.7 | 48.8 | 28.9 | 35.4 |
| | IPSeg (ours) | Swin-B | 43.2 | 30.4 | 38.9 | 43.0 | 30.9 | 39.0 | 43.8 | 31.5 | 39.7 | 51.1 | 34.8 | 40.3 |

due to catastrophic forgetting. This detailed trend of performance decline across steps validates the effectiveness and robustness of IPSeg in resisting forgetting.

**Results on ADE20K**  We also perform a comparison between IPSeg and its competitors under different incremental scenarios on the more challenging ADE20K dataset with two backbones. As shown in Table 2, IPSeg consistently achieves performance advantages similar to those observed on Pascal VOC 2012. Notably, IPSeg with Swin-B backbone demonstrates more significant improvements over its competitors across all incremental scenarios on the ADE20K dataset, with the smallest improvement of **2.1**% in the 100-5 scenario and the largest improvement of **6.0**% in the 100-50 scenario. The superior performance on the more realistic and complex ADE20K dataset further demonstrates the effectiveness and robustness of IPSeg.

**Qualitative analysis of IPSeg**  Figure 4 presents a qualitative analysis of IPSeg compared with SSUL-M and CoinSeg-M. Visualization results are from each incremental step in the VOC 2-2 scenario. The results in rows 1, 3, and 5 demonstrate that both SSUL-M and CoinSeg-M mistakenly predict "horse" as "cow" at step 6, while IPSeg correctly identifies "horse". In rows 2, 4, and 6, IPSeg consistently predicts the old class "chair", whereas SSUL-M predicts "sofa" as "table" at step 6, and both SSUL-M and CoinSeg-M mistake "chair" for "sofa" at step 8. These visualization results reveal that IPSeg not only achieves excellent learning plasticity but also maintains strong memory stability.

### 4.3 ABLATION STUDY

**Ablation study on IPSeg**  We analyze the effect of the proposed components in IPSeg, including Image Posterior (**IP**), Semantics Decoupling (**SD**), and the design of Noise Filtering (**NF**) in SD. All

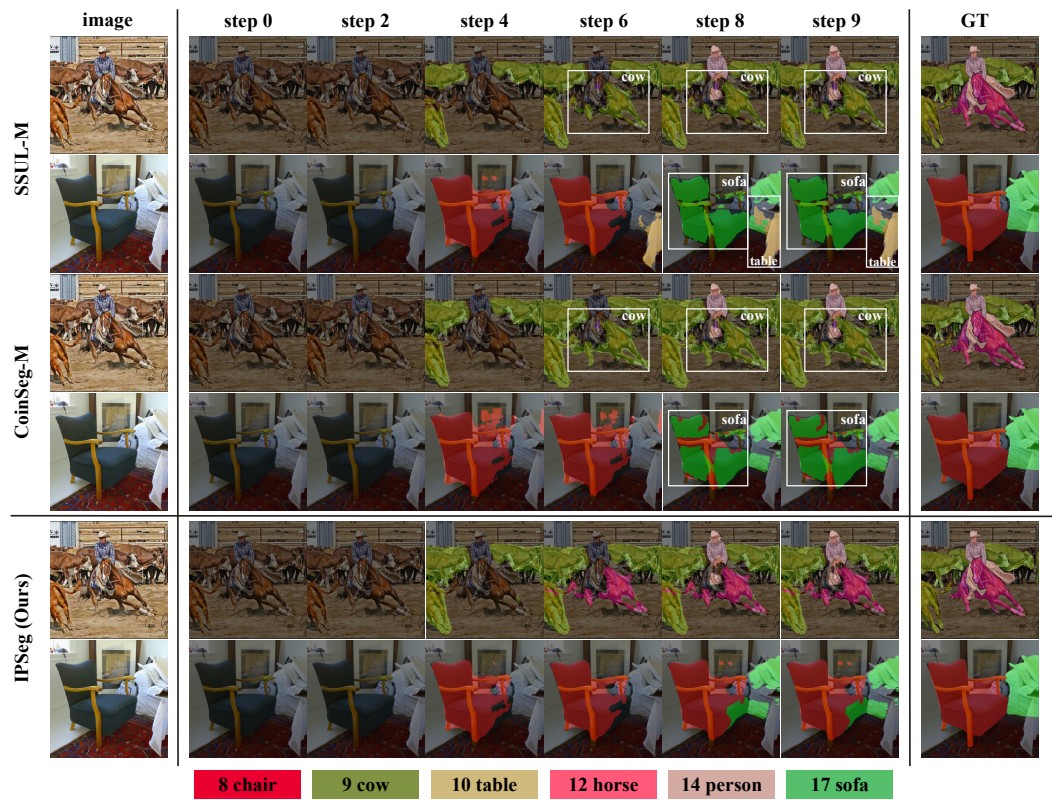

Figure 4: Qualitative analysis of IPSeg on Pascal VOC 2012. Texts and bounding boxes in white indicate incorrect class predictions starting from the corresponding incremental step. Color-shaded boxes with classes and indexes represent the learned classes in the corresponding learning steps.

Table 3: (a): Overall ablation study for IPSeg on VOC 15-1. (b) and (c): Ablation studies for hyper-parameters: memory size $|M|$, the ratio of noise filtering $\alpha_{NF}$ and the ratio of background compensation $\alpha_{BC}$.

| IP | SD | NF | VOC 15-1 (6 steps) | | |
|---|---|---|---|---|---|
| | | | 0-15 | 16-20 | all |
| ✗ | ✗ | ✗ | 78.8 | 49.7 | 71.9 |
| ✓ | ✗ | ✗ | 79.4 | 69.6 | 77.0 |
| ✗ | ✓ | ✗ | 83.1 | 65.1 | 78.8 |
| ✗ | ✓ | ✓ | 83.4 | 69.6 | 80.1 |
| ✓ | ✓ | ✗ | 83.4 | 74.7 | 81.3 |
| ✓ | ✓ | ✓ | **83.6** | **75.1** | **81.6** |

(a)

| | value | 0-15 | 16-20 | all |
|---|---|---|---|---|
| $|M|$ | 50 | 83.4 | 72.3 | 80.8 |
| | 100 | 83.5 | 75.1 | 81.5 |
| | 200 | 83.5 | 75.5 | 81.7 |
| $\alpha_{NF}$ | 0.2 | 83.5 | 75.0 | 81.4 |
| | 0.4 | **83.6** | **75.1** | **81.6** |
| | 0.6 | 83.4 | 74.6 | 81.3 |
| | 0.8 | 83.3 | 74.2 | 81.2 |
| | 1.0 | 83.4 | 74.7 | 81.3 |

(b)

| $\alpha_{BC}$ | VOC 15-1 (6 steps) | | |
|---|---|---|---|
| | 0-15 | 16-20 | all |
| 1 | 82.9 | 74.9 | 81.0 |
| 0.9 | **83.5** | **75.1** | **81.6** |
| 0.8 | 83.5 | 75.0 | 81.5 |
| 0.7 | 83.4 | 74.4 | 81.3 |
| 0.6 | 83.3 | 74.3 | 81.2 |
| 0.5 | 83.2 | 73.8 | 81.0 |
| 0 | 80.6 | 66.6 | 77.3 |

(c)

ablations are implemented under the 15-1 setting of Pascal VOC 2012 using the Swin-B backbone. As shown in Table 3(a), the second row indicates that **IP** brings significant improvement to new classes. Benefiting from **IP**'s ability to maintain consistency between tasks, the reliable guidance effectively prevents model performance from degradation caused by *separate optimization*. The third row illustrates the excellent ability of **SD** in learning foregrounds at each step. **SD** consists of a permanent branch and a temporary branch, and decouples noisy information into static and dynamic groups, allowing the model to individually learn the corresponding knowledge. The fifth row demonstrates IPSeg's outstanding performance on both old and new classes with **IP** and **SD** together. The last row shows that **NF**, as an additional trick, further enhances performance.

**Ablation study of hyper-parameters**   Table 3(b) and Table 3(c) illustrate the effects of hyper-parameters: memory size $|\mathcal{M}|$, the strength of noise filtering $\alpha_{NF}$, and background compensation $\alpha_{BC}$,

Table 4: Comparison of different label choices to incrementally train the image posterior branch. The choice of Pseudo is adopted in IPSeg.

| Methods | Labels $\mathcal{C}_{1:t-1}$ | $\mathcal{C}_t$ | VOC 15-5 (2 steps) 0-15 | 16-20 | all | VOC 10-1 (11 steps) 0-10 | 11-20 | all | VOC 2-2 (10 steps) 0-2 | 3-20 | all |
|---|---|---|---|---|---|---|---|---|---|---|---|
| Part-GT | ✗ | GT | 83.3 | 72.8 | 80.8 | 79.3 | 74.5 | 77.0 | 72.6 | 69.4 | 69.8 |
| Pseudo (ours) | PL | GT | 83.3 | 73.3 | 80.9 | 80.3 | 76.7 | 78.6 | 73.1 | 72.3 | 72.4 |
| Full-GT | GT | GT | 83.2 | 73.8 | 81.0 | 80.1 | 78.0 | 79.2 | 75.2 | 74.6 | 74.8 |

Table 5: The ablation study of **SD** over background (BG), base foreground classes (Base) and new foreground classes (New) on Pascal VOC 15-1 with Swin-B backbone.

| SD | BG | Base(1-15) | New(16-20) | All |
|---|---|---|---|---|
| ✗ | 92.4 | 78.5 | 69.6 | 77.0 |
| ✓ | 94.3(+1.9) | 82.3(+3.7) | 75.1(**+5.5**) | 81.5 |

which shows that IPseg is not sensitive to the value of $\alpha_{\text{NF}}$ and $\alpha_{\text{BC}}$ and we set the default values for these parameters to $|\mathcal{M}| = 100$, $\alpha_{\text{NF}} = 0.4$ and $\alpha_{\text{BR}} = 0.9$. Besides, we set loss trade-off factors $\lambda_1 = \lambda_2 = 0.5$ in our implementation, and more details are provided in the appendix.

**Ablation study on image-level pseudo-label $\tilde{\mathcal{Y}}_k$** The image posterior (IP) branch is trained incrementally but faces challenges due to the lack of labels for old classes. To address this issue, we employ image-level pseudo-label $\tilde{\mathcal{Y}}_k$ (PL) instead of directly using the partial ground truth label $\mathcal{Y}_k$, providing richer supervision at the risk of introducing some noise due to the inconsistencies between previous heads predictions and current dataset labels. As shown in Table 4, our method achieves significant improvement compared to using only partial ground truth (Part-GT), and further improvements can be observed when using full ground truth (Full-GT). This indicates that using $\tilde{\mathcal{Y}}_k$ is an efficient trade-off, where the benefits of additional supervision outweigh the potential noise.

**Impact of semantics decoupling on temporary concepts** To understand which types of concepts most benefit from IPSeg, we categorize the 20 classes of Pascal VOC into 15 base classes and 5 new classes based on the incremental process. According to the learning objectives, all foreground classes are treated as temporary concepts in the corresponding step and the background is constantly considered as permanent ones. The comparison shown in Table 5 indicates that the new classes gain more significant performance improvement than the base classes. Furthermore, the permanent concepts (i.e., the background) achieve less improvement compared to the temporary concepts. This observation suggests that IPSeg is more effective in enhancing the learning of new foreground classes. The detailed class-wise results are provided in the appendix.

## 5 CONCLUSIONS

In this paper, we propose IPSeg, a simple yet effective method designed to address the issue of semantic drift in class incremental semantic segmentation. We begin by analyzing the details of semantic drift, identifying two key issues: separate optimization and noisy semantics. To mitigate these issues, IPSeg introduces two specific designs: image posterior guidance and semantics decoupling. Experimental results on the Pascal VOC 2012 and ADE20K datasets demonstrate the superior performance of our method, particularly in long-term incremental scenarios. Moreover, IPSeg exhibits excellent properties in terms of both learning plasticity and memory stability.

**Limitations and social impact** Though IPSeg exhibits superior performance and properties of learning plasticity and memory stability, we acknowledge that the usage of memory buffers leaves a lot of room for discussion. On the one hand, the use of memory buffers brings additional storage costs and the risk of information leakage. On the other hand, the use of memory buffers is related to privacy issues, such as storing private information without approval. These issues need to be treated with caution in artificial intelligence applications.

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

# A APPENDIX

## A.1 SYMBOLS AND EXPLANATIONS

Table 6 provides key symbols used in our paper along with their explanations to facilitate a better understanding.

Table 6: Symbols and explanations

| | Symbol | Explanations |
|---|---|---|
| Model architecture | $f_t$ | Model of task $t$. |
| | $h_\theta$ | The backbone extracting features. |
| | $\phi_{1:t}$ | All segmentation heads, outputting the final results. |
| | $\phi_t$ | The head of task $t$, representing the temporary branch. |
| | $\phi_p$ | The permanent branch. |
| | $\psi$ | The image posterior branch. |
| Semantic concepts | $\mathcal{C}_t$ | The target classes set of task $t$. |
| | $\mathcal{C}_{1:t-1}$ | The old classes set. |
| | $c'_b$ | The pure background. |
| | $c'_u$ | Unknown foreground. |
| | $c_i$ | The ignored region that does not participate in loss calculation. |
| | $c_f$ | The foreground regions that do not belong to target classes $\mathcal{C}_t$. |
| | $c_u$ | Unknown classes defined in previous methods, consisting of $\mathcal{C}_{1:t-1}$ and $c'_u$. |
| | $c_b$ | The background defined in previous methods, consisting of $c'_b$ and $c_u$. |
| Data and label | $\mathcal{D}_t$ | Training dataset of current incremental task $t$. |
| | $\mathcal{M}$ | Memory buffer, with fixed size of 100 for Pascal VOC and 300 for ADE20K. |
| | $x_i^t$ | The $i$-th image in $\mathcal{D}_t$. |
| | $y_i^t$ | The pixel annotations of $x_i^t$. |
| | $x_i^{m,t}$ | The mixed data selected from $\mathcal{M}$ and $\mathcal{D}_t$. |
| | $\tilde{\mathcal{Y}}_i^t$ | The image-level pseudo-label of $x_i^{m,t}$. |
| | $\tilde{y}_i^p$ | Label assigned to the permanent branch $\phi_p$. |
| | $\tilde{y}_i^t$ | Label assigned to the temporary branch $\phi_t$. |

## A.2 OBSERVATION

**Visualization of separate optimization** As shown in Figure 5, to illustrate the inconsistent outputs caused by *separate optimization*, we select three pairs of similar classes from Pascal VOC 2012: "cow" and "horse", "bus" and "car", "sofa" and "chair", and split them into two separate groups for learning. Sequence B follows a reverse learning order compared to Sequence A, and the goal is to examine the model's final predictions with different learning sequence. Columns A and B are the models' final predictions of sequence A and sequence B respectively. These visualizations indicate that the earlier head of the model tends to produce high scores for certain classes, regardless of the learning order, suggesting that *separate optimization* causes a persistent bias towards the classes learned first.

**Impact of IPSeg on separate optimization** To validate the impact of IPSeg on *separate optimization*, we calculate the average probability distribution of the incorrect prediction area (red box in the image) as depicted in Figure 6. SSUL-M misclassifies the little sheep as cow with abnormal probability distribution. In contrast, IPSeg utilizes image posterior guidance to produce more accurate

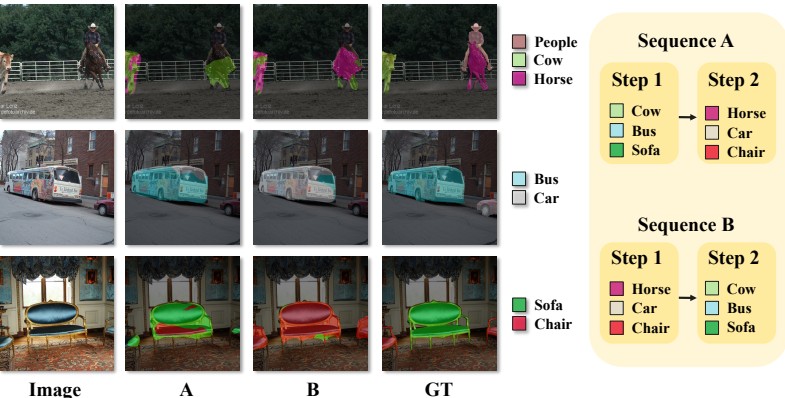

Figure 5: The visualization of separate optimization. *Sequence A*: first learn "cow", "bus" and "sofa" in step 1, then "horse", "car" and "chair" in step 2. *Sequence B*: first learn "horse", "car" and "chair" in step 1, then "cow", "bus" and "sofa" in step 2.

and harmonious prediction. Compared to previous works, IPseg maintains a harmonious and realistic probability distribution more similar to that of the theoretical upper bound, Joint-Training (Joint), demonstrating its superior capability in dealing with *separate optimization*.

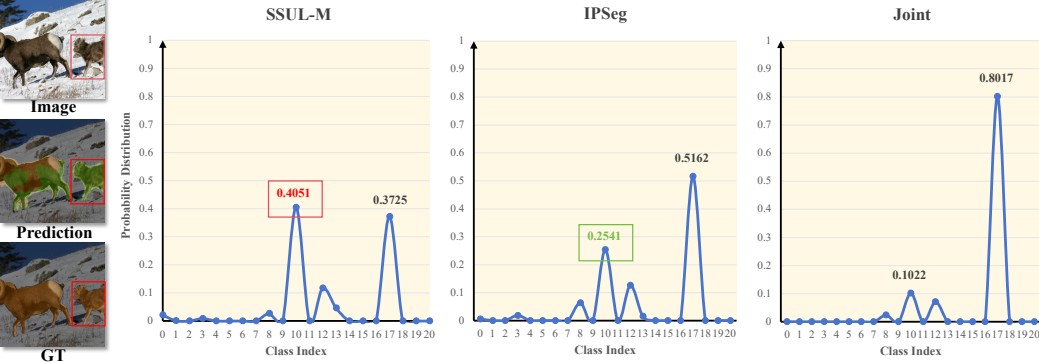

Figure 6: The probability distributions for SSUL-M, IPSeg, and Joint-Training (Joint) in the regions of incorrect predictions. Class indexes "10" and "17" represent "cow" and "sheep" respectively.

**Visualization of semantics decoupling** Figure 7 is the semantics decoupling illustration for image $x_i^t$. In this case, the current classes $\mathcal{C}_t$ "person" is provided with ground truth as $y_i^t$. The foreground classes "sofa" and "cat", however, are unknown without ground truth. IPSeg uses a saliency map to locate the current unknown object "sofa" and "cat" as other foregrounds $c_f$ and further utilizes pseudo label to distinguish "sofa" as unknown foreground $c_u'$. It is worth noting that the regions of "person" and "cat" belong to the ignored regions $c_i$ that do not participate in loss calculation. In this way, the remaining region is labeled as "pure" background $c_b'$. The "pure" background $c_b'$ and unknown foreground $c_u'$ are considered as static and permanent concepts. The target classes $\mathcal{C}_t$ with ground truth $y_i^t$ and other foreground $c_f$ are considered as dynamic and temporary concepts.

### A.3 ADDITIONAL RESULTS AND ANALYSIS

**Evaluation on image-level predictions** To investigate the ability to resist catastrophic forgetting of the image posterior (IP) branch and the segmentation branch, we evaluate the image level accuracy performance of the base 15 classes using the IP branch and the segmentation branch at each step on Pascal VOC 15-1 as shown in Table 7. "IP" refers to the image-level accuracy of the IP branch, "Pixel" refers to the image-level accuracy of the segmentation branch, where class $\mathcal{C}$ exists if a pixel is

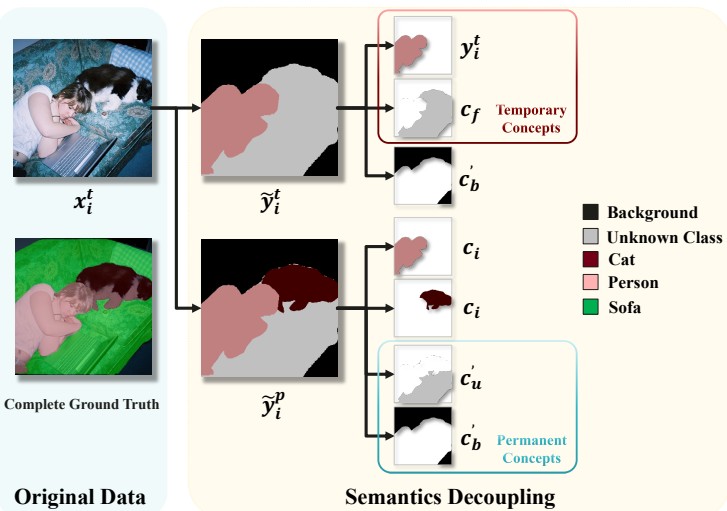

Figure 7: Semantics decoupling strategy

predicted as $\mathcal{C}$. "Pixel+IP" denotes the final result of IPSeg. The ablation shows that: the image-level performance suffers less forgetting than the segmentation, and our method shows similar property against forgetting with the help of IP.

Table 7: The image-level accuracy of IP branch and the segmentation branch on 15 base classes.

| ACC (%) | step 0 | step 1 | step 2 | step 3 | step 4 | step 5 |
|---|---|---|---|---|---|---|
| IP | 87.44 | 86.41 | 86.99 | 86.82 | 86.86 | 86.29 |
| Pixel | 88.17 | 86.42 | 86.30 | 85.43 | 84.84 | 84.70 |
| Pixel+IP | 93.07 | 92.24 | 92.41 | 91.93 | 91.95 | 91.02 |

We also present the image-level accuracy on all seen classes at each step to analyze their performance on both old classes and new classes in Table 8. For the segmentation branch, the image-level accuracy of it on all seen classes gradually degrades after learning new classes, performing worse than its accuracy on base classes. This indicates the segmentation branch performs poorly on new classes, which is consistent with our description about separate optimization. In contrast, the IP branch experiences less deterioration from separate optimization and help our method maintain a good balance between retaining old knowledge and learning new knowledge.

Table 8: The image-level accuracy of IP branch and the segmentation branch on all seen classes.

| ACC (%) | step 0 | step 1 | step 2 | step 3 | step 4 | step 5 (Final) |
|---|---|---|---|---|---|---|
| IP | 87.44 | 82.54 | 81.14 | 81.32 | 82.09 | **82.34** |
| Pixel | 88.17 | 83.56 | 82.29 | 78.23 | 77.60 | **76.57** |
| Pixel+IP | 93.07 | 90.05 | 90.13 | 87.30 | 87.68 | **88.03** |

**Evaluation on image posterior branch**   Image posterior (IP) branch is trainable during different incremental steps, and it will undoubtedly suffer from catastrophic forgetting. To evaluate the effectiveness of IP branch, we conduct a quantitative evaluation of it as shown in Table 9. We train IP branch with a multi-label classification task in three scenarios and report the precision, recall, and F1 results: "Seq" refers to training the IP branch sequentially without any additional tricks, indicating the worst case suffering from catastrophic forgetting, "Ours" refers to the training setting used in our paper, and "Joint" refers to training with all task data jointly serving as the upper bound. It is obvious that IP branch suffers little forgetting. We achieve this performance mainly owing to two specific settings:

- **Mixed training data:** IP branch uses data from both the memory buffer and the current task dataset for training. Compared to fine-grained task heads, it is easier for IP branch to learn image-level knowledge from memory buffer.

- **Image-level pseudo-label for supervision:** IPSeg introduces image-level pseudo-label on past classes as supervision to mitigate catastrophic forgetting challenge. The ablation result in Table 4 partially reflects the effectiveness of this design.

Table 9: The performance of image posterior branch on different settings.

| Method | Precision (%) | Recall (%) | F1 (%) |
|--------|--------------|-----------|--------|
| Seq    | 55.62        | 24.67     | 23.78  |
| Ours   | 78.28        | 87.03     | 80.68  |
| Joint  | 89.96        | 90.00     | 89.89  |

**Evaluation on model parameters, training and inference costs**    We provide a comprehensive analysis of the model parameters, training, and inference costs as shown in Table 10. We test and report the results of IPSeg, SSUL-M and CoinSeg-M with Swin-B on the VOC 15-1 setting. We set *image_size=512x512, epochs=50, and batch_size=16* in training and *image_size=512x512* for inference test. All results are run on RTX 3090 GPU.

- **Model Parameters:** Using the thop tool, we analyze and compare the trainable parameters for these methods. The sizes of increased parameters in them are close, with average $3.84M$ per step. Additionally, IPSeg has $29.72M$ parameters more than SSUL due to the additional image posterior branch.

- **Training:** Due to the introduced image posterior branch, IPSeg needs more training cost compared with SSUL-M but less than CoinSeg-M.

- **Inference:** The inference speed of IPSeg (27.3 FPS) is slightly lower than SSUL-M (33.7 FPS) and similar to CoinSeg-M (28.2 FPS). Due to the proposed image posterior branch, the model's floating-point operations (137.1 GFLOPs) are higher than the baseline (94.9 GFLOPs), and with an approximately 1 GB increase in GPU usage.

Overall, IPSeg introduces an additional image posterior branch with slight increases in model parameters, training cost, and inference but brings great performance improvement. It is a worthwhile trade-off between performance and cost.

Table 10: Comparison of IPSeg with baseline on model parameters, training and inference costs.

| Method | Incremental Steps | | | | | | Training | | Inference | | |
|--------|---|---|---|---|---|---|------|-----------|-----|-------|-----------|
|        | 0 | 1 | 2 | 3 | 4 | 5 | Time | GPU usage | FPS | FLOPs | GPU usage |
| IPseg    | 135.92 M | 139.76 M | 143.60 M | 147.66 M | 151.28 M | 155.12 M | 9h 14min | 21.1G | 27.3 | 137.1G | 6.2G |
| SSUL-M   | 106.20 M | 110.03 M | 113.89 M | 117.95 M | 121.56 M | 125.40 M | 7h 13min | 19.4G | 33.7 | 94.9G  | 5.3G |
| CoinSeg-M | 107.02 M | 111.15 M | 115.29 M | 119.42 M | 123.55 M | 127.68 M | > 15h   | 21.3G | 28.2 | 96.3G  | 5.6G |

**Ablation study on salient map**    The salient map is effective for Pascal VOC 2012, but not as useful for ADE20K as we expected. Specifically, the use of saliency map on ADE20K presents challenges in correctly identifying target regions, especially for objects that are not at the center of the image and cover large areas. To evaluate the quality of salient map, we conduct ablation on the ADE20K 100-10 task with the following settings as shown in Table 11: "w/o Sal" uses no saliency map supervision, "w/ Sal" uses saliency map supervision as we do in paper, and "w/ SAM" uses saliency map extracted by SAM model. We use the official Grounded SAM (Ren et al., 2024; Kirillov et al., 2023) code with all classes in ADE20K used as text prompts to extract corresponding masks. Additionally, we also report performance differences on "Thing" and "Stuff" classes defined in ADE20K panoptic segmentation (Zhou et al., 2019) to investigate the bias of saliency map on different semantic regions. Basically, we find the following conclusion:

- Using the default saliency map supervision to implement knowledge decoupling strategy, IPSeg gets a performance improvement by +1.6% mIoU. And using saliency extracted by SAM further improves IPSeg performance by 0.7% mIoU.

- The default saliency maps perform well in identifying "Things" classes but struggle with "Stuff" classes, resulting in a performance gain of +1.9% on "Things" classes but merely +0.1% on "Stuff" classes. Furthermore, the SAM-based saliency maps provide better supervision for both "Things" and "Stuff" classes, with improvements of +0.6% on "Things" and +1.1% on "Stuff" compared to "w/ Sal".

Table 11: Ablation study on salient map of IPSeg on ADE20K 100-10 tasks.

| Method | 0-100 | 101-150 | all | Things | Stuff |
|---|---|---|---|---|---|
| w/o Sal | 42.1 | 28.0 | 37.4 | 37.2 | 37.5 |
| w/ Sal | 43.0 | 30.9 | 39.0 | 39.1 | 37.6 |
| w/ SAM | 43.6 | 31.8 | 39.7 | 39.7 | 38.7 |

**The details of efficient data storage in memory buffer**  For raw data, IPSeg directly stores the image paths in a JSON file, as done in previous works (Cha et al., 2021; Zhang et al., 2022b; 2023). For image-level labels, IPSeg stores the class labels of the images as arrays in the same JSON file with multi-hot encoding, where 1 indicates the presence of a class and 0 indicates absence. The memory cost for this is negligible. For pixel-level labels, instead of storing full-class annotations (with a data type of *uint8* ) as prior approaches, IPSeg only stores the salient mask, where the background and foreground are labeled as 0 and 1, respectively (with a data type of *bool* ). Theoretically, the storage space could be reduced to $1/8$.

**Ablation study on memory buffer**  Since IPSeg is designed for data-replay scenarios, the IP branch heavily relies on a memory buffer. To evaluate the impact of the memory buffer on performance, we compare the standard version of IPSeg with a data-free variant (denoted as IPSeg w/o M). As shown in Table 12, IPSeg demonstrates competitive performance even without the memory buffer. However, the performance gap between the data-free and data-replay settings highlights the essential role of the memory buffer in enhancing IPSeg's effectiveness.

Table 12: Comparison with other methods in data-free version using Swin-B backbone.

| Method | VOC 15-5 (2 steps) | | | VOC 15-1 (6 steps) | | | VOC 10-1 (11 steps) | | | VOC 2-2 (10 steps) | | |
|---|---|---|---|---|---|---|---|---|---|---|---|---|
| | 0-15 | 16-20 | all | 0-15 | 16-20 | all | 0-10 | 11-20 | all | 0-2 | 3-20 | all |
| SSUL | 79.7 | 55.3 | 73.9 | 78.1 | 33.4 | 67.5 | 74.3 | 51.0 | 63.2 | 60.3 | 40.6 | 44.0 |
| MicroSeg | **81.9** | 54.0 | 75.2 | 80.5 | 40.8 | 71.0 | 73.5 | 53.0 | 63.8 | 64.8 | 43.4 | 46.5 |
| IPSeg w/o M | 81.4 | **62.4** | **76.9** | **82.4** | **52.9** | **75.4** | **80.0** | **61.2** | **71.0** | **72.1** | **64.5** | **65.5** |
| IPSeg w/ M | 83.3 | 73.3 | 80.9 | 83.5 | 75.1 | 81.5 | 80.3 | 76.7 | 78.6 | 73.1 | 72.3 | 72.4 |

**Ablation study for hyper-parameters: weight terms of loss.**  We conduct an ablation study on the two weight terms $\lambda_1$ and $\lambda_2$, testing values of 0.1, 0.25, 0.5, 0.75, and 1.0. The results are shown in Table 13. It is obvious that the setting of $\lambda_1 = 0.5$ and $\lambda_2 = 0.5$ achieves the best performance, which is the default value of IPSeg.

**Detailed results of semantics decoupling on temporary concepts**  To understand which types of classes or concepts benefit from our method, we compare IPseg against the baseline on 20 classes of the VOC 15-1 setting. The detailed results are presented in Table 14, which demonstrates that IPSeg is more effective in enhancing the learning of new foreground classes.

**Class-wise results of IPSeg**  Table 15 shows the detailed experimental results of IPSeg for each class across four incremental scenarios of Pascal VOC 2012. IPSeg demonstrates superior performance in various incremental learning tasks, including standard tasks with a large number of initial classes (e.g., 15-5 and 15-1) and long-range tasks with fewer initial classes (e.g., 10-1 and 2-2). Notably, in the 2-2 task, the mIoU for "cow" and "horse" reaches 70.7% and 81.4%, respectively. This indicates

Table 13: Ablation Studies on Pascal VOC 15-1 task for hyper-parameters: weight terms of loss, $\lambda_1$ and $\lambda_2$.

| $\lambda_1 \setminus \lambda_2$ | 0.1 | 0.25 | 0.5 | 0.75 | 1.0 |
|---|---|---|---|---|---|
| 0.1 | 79.2 | 80.3 | 80.4 | 80.8 | 78.0 |
| 0.25 | 79.8 | 80.3 | 81.1 | 81.0 | 80.9 |
| 0.5 | 80.3 | 80.9 | **81.5** | 81.2 | 80.9 |
| 0.75 | 81.1 | 81.3 | 81.1 | 81.0 | 81.0 |
| 1.0 | 80.8 | 81.3 | 81.1 | 81.2 | 80.1 |

Table 14: Detailed results of the ablation study for semantics decoupling (**SD**) over each class on Pascal VOC 15-1 with Swin-B backbone. Texts in red indicate 5 new classes.

| SD | | Detailed results | | | | | | | | | |
|---|---|---|---|---|---|---|---|---|---|---|---|
| ✗ | BG | plane | bike | bird | boat | bottle | bus | car | cat | chair | cow |
| | 92.4 | 87.4 | 37.7 | 89.1 | 67.8 | 80.4 | 93.8 | 86.9 | 93.6 | 43.9 | 85.7 |
| | tabel | dog | horse | motor | person | plant | sheep | sofa | train | TV | **mIoU** |
| | 63.8 | 90.0 | 87.2 | 85.9 | 84.1 | 57.5 | 81.6 | 53.3 | 87.1 | 68.4 | **77.0** |
| ✓ | BG | plane | bike | bird | boat | bottle | bus | car | cat | chair | cow |
| | 94.3 | 91.8 | 43.8 | 93.8 | 75.0 | 86.0 | 94.2 | 91.2 | 96.1 | 44.3 | 94.6 |
| | tabel | dog | horse | motor | person | plant | sheep | sofa | train | TV | **mIoU** |
| | 67.3 | 94.5 | 93.0 | 88.8 | 88.2 | 65.6 | 90.1 | 57.9 | 89.3 | 72.7 | **81.5** |

that IPSeg maintains excellent prediction performance even when classes with similar semantic information are trained in different stages.

Table 15: Class-wise results of IPSeg over each class.

| | BG | plane | bike | bird | boat | bottle | bus | car | cat | chair | cow |
|---|---|---|---|---|---|---|---|---|---|---|---|
| **VOC 15-5** | 93.6 | 92.2 | 44.9 | 93.8 | 74.4 | 85.2 | 93.9 | 90.8 | 96.2 | 43.0 | 94.5 |
| | tabel | dog | horse | motor | person | plant | sheep | sofa | train | TV | **mIoU** |
| | 68.0 | 94.2 | 92.9 | 88.0 | 88.0 | 66.3 | 91.5 | 46.4 | 87.8 | 74.4 | **80.9** |
| **VOC 15-1** | BG | plane | bike | bird | boat | bottle | bus | car | cat | chair | cow |
| | 94.3 | 91.8 | 43.8 | 93.8 | 75.0 | 86.0 | 94.2 | 91.2 | 96.1 | 44.4 | 93.5 |
| | tabel | dog | horse | motor | person | plant | sheep | sofa | train | TV | **mIoU** |
| | 67.4 | 94.5 | 92.9 | 88.8 | 88.2 | 66.4 | 88.6 | 58.1 | 89.5 | 72.7 | **81.5** |
| **VOC 10-1** | BG | plane | bike | bird | boat | bottle | bus | car | cat | chair | cow |
| | 93.1 | 93.1 | 42.2 | 93.2 | 72.1 | 83.5 | 93.9 | 91.9 | 95.8 | 38.0 | 86.9 |
| | tabel | dog | horse | motor | person | plant | sheep | sofa | train | TV | **mIoU** |
| | 54.1 | 91.8 | 86.1 | 87.5 | 87.5 | 64.2 | 85.6 | 49.9 | 88.4 | 72.1 | **78.6** |
| **VOC 2-2** | BG | plane | bike | bird | boat | bottle | bus | car | cat | chair | cow |
| | 91.4 | 88.6 | 39.3 | 87.9 | 71.5 | 71.9 | 89.1 | 78.2 | 89.3 | 28.8 | 70.7 |
| | tabel | dog | horse | motor | person | plant | sheep | sofa | train | TV | **mIoU** |
| | 55.2 | 83.5 | 84.1 | 77.6 | 82.6 | 63.6 | 72.1 | 44.4 | 82.1 | 69.1 | **72.4** |

**Experimental results of disjoint setting**   To demonstrate IPSeg's robust learning capability under different incremental learning settings and to further prove its superiority over general methods, we evaluate IPSeg using the disjoint setting on the Pascal VOC 2012 dataset for the 15-1 and 15-5 tasks, as shown in Table 16. The results indicate that IPSeg consistently achieves the best performance compared to state-of-the-art methods. Additionally, similar to the results in the overlap setting, IPSeg exhibits a strong ability to learn new classes while retaining knowledge of the old classes. Specifically, IPSeg outperformed the second-best method by **10.1**% in the 15-5 task and by **22.8**% in the 15-1 task in terms of new class performance.

Table 16: Comparison with state-of-the-art methods on Pascal VOC 2012 dataset for disjoint setup.

| | Method | VOC 15-5 (2 steps) | | | VOC 15-1 (6 steps) | | |
|---|---|---|---|---|---|---|---|
| | | 0-15 | 16-20 | all | 0-15 | 16-20 | all |
| Data-free | LwF-MC | 67.2 | 41.2 | 60.7 | 4.5 | 7.0 | 5.2 |
| | ILT | 63.2 | 39.5 | 57.3 | 3.7 | 5.7 | 4.2 |
| | MiB | 71.8 | 43.3 | 64.7 | 46.2 | 12.9 | 37.9 |
| | RCIL | 75.0 | 42.8 | 67.3 | 66.1 | 18.2 | 54.7 |
| Replay-based | SDR | 74.6 | 44.1 | 67.3 | 59.4 | 14.3 | 48.7 |
| | SSUL-M | 76.5 | 48.6 | 69.8 | 76.5 | 43.4 | 68.6 |
| | MicroSeg-M | 80.7 | 55.2 | 74.7 | 80.0 | 47.6 | 72.3 |
| | CoinSeg-M | 82.9 | 61.7 | 77.9 | 82.0 | 49.6 | 74.3 |
| | IPSeg(ours) | **82.7** | **71.8** | **80.1** | **82.6** | **72.4** | **80.2** |

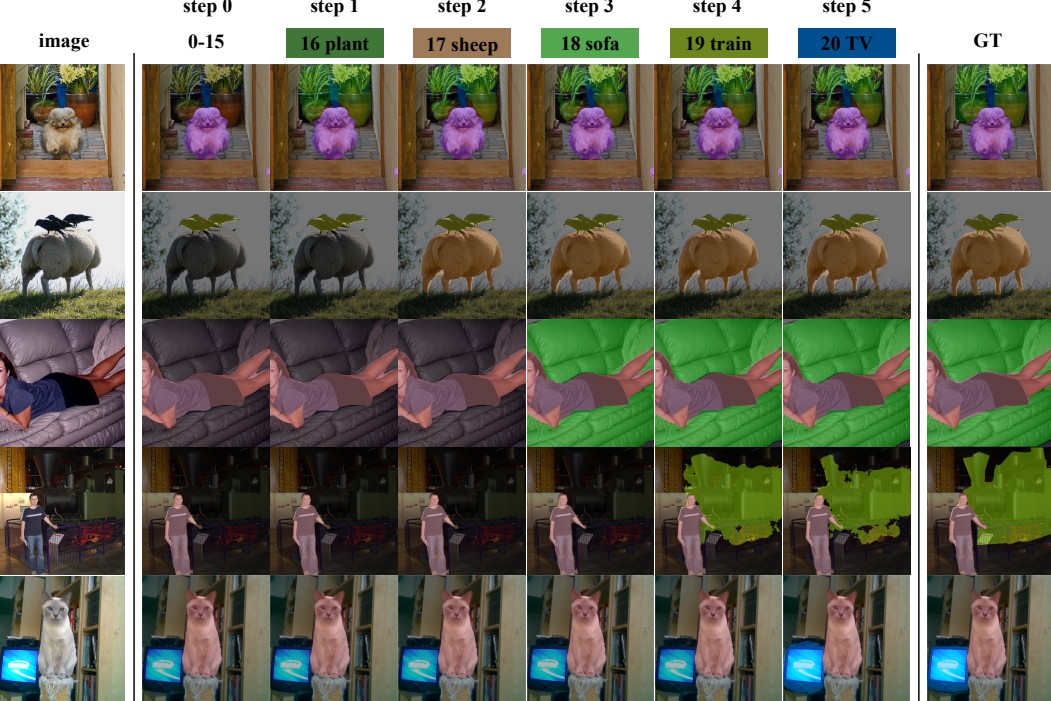

Figure 8: Qualitative results on Pascal VOC 2012 dataset with the 15-1 scenario.

**More qualitative results on Pascal VOC 2012**   In addition to the qualitative results of the VOC 2-2 task shown in the main paper, we present additional qualitative analysis in Figure 8. We select the 15-1 task and perform a visual analysis for each newly added class. Each image includes both old and new classes, covering indoor and outdoor scenes as well as various objects and environments. For example, the first row shows the learning ability for "plant". After step 1, the model consistently predicts "plant" correctly while retaining the ability to recognize "dog". Similarly, rows 2-5 show consistent performance. For each new class (i.e., sheep, sofa, train, and TV in the figure), IPSeg quickly adapts to them while retaining the ability to recognize old classes (i.e., bird, person, cat in the figure). These results clearly and intuitively demonstrate IPSeg's strong capability in addressing incremental tasks.

**More qualitative results on ADE20K**   The qualitative results of the 100-10 task on the ADE20K dataset are shown in Figure 9. We select five examples to illustrate the model's ability to predict various classes as the learning step increases. Row 1 shows the performance of predicting the new class "ship" in step 1, where the model effectively recognizes both the old class "sky" and the new class "ship." Similarly, in rows 2-5, for the newly introduced classes (tent, oven, screen, flag), IPSeg

demonstrates excellent performance in learning the new classes without forgetting the old ones. This indicates that IPSeg achieves a balance between stability and plasticity even on more challenging and realistic datasets.

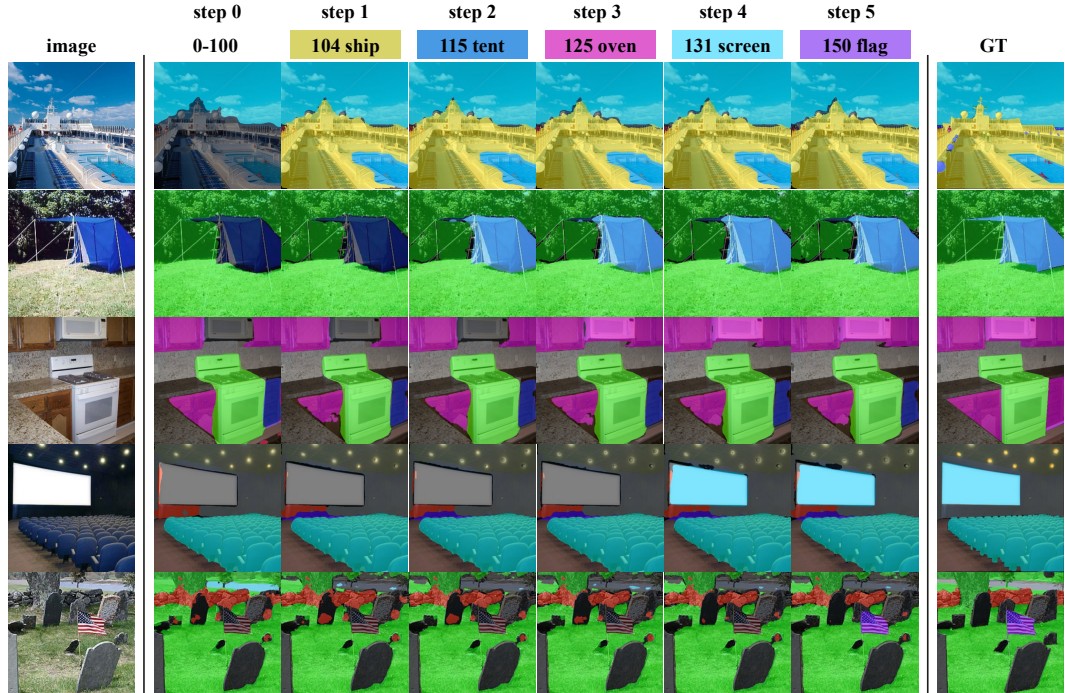

Figure 9: Qualitative results on ADE20K dataset with the 100-10 scenario.

