# OpenReview forum: "IPSeg: Image Posterior Mitigates Semantic Drift in Class-Incremental Segmentation"
_ICLR.cc/2025/Conference — Submitted to ICLR 2025_

### Official Review · Reviewer_FU4w · 2024-11-02

**Soundness:** 3
**Presentation:** 3
**Contribution:** 3
**Rating:** 5
**Confidence:** 4

**Summary:**

To address the challenging Class incremental semantic segmentation (CISS), this paper proposes Image Posterior to attack the noisy semantics and Semantic Decoupling to cope with separate optimization, which both of these two issues refer to semantic drift and background shift problems in CISS. The proposed Image posterior supports with guidance to calibrate the error prediction and amplify the scale of correct prediction, hence solving the semantic drift issue when the model faces ambiguous categories between the past task and the current task. Semantic Decoupling implemented by Memory Bank manner is used to decouple the pure background, unknown foreground, past class pixels, and target class pixels. The authors employ a saliency estimator and filtering trick to improve the memory buffer while the memory buffer is an image-level corpus.

**Strengths:**

1. The paper writing is kind of good.

2. Supporting image posterior within the image-level to guide the CISS task sounds reasonable to resolve the semantic drift and works like error calibration.

3. The final CISS experimental results seem competitive and the ablative studies are extensive.

**Weaknesses:**

1. Decoupling the background into pure background and unknown foregrounds (utilizing the saliency model), I believe, has been widely studied. The decoupling implementation sounds complicated to understand in the main paper, but I think the authors could provide a more clear presentation of this part. I think the authors should make detailed comparisons with existing works, like Ssul.

2. Not sure how the authors construct the memory buffer with image-level samples. This plays an important role in this paper to obtain good guidance.

3. The figure presentations are inconsistent, Figure shows good quality while figures 2/4/5/6/7/8 display a blur effect. These should be presented by pdf, etc.

**Questions:**

Please refer to my weaknesses part. Moreover, the authors shall further make some comparisons in terms of training efficacy with the previous method, since they use a memory buff and constitute a two-branches architecture.

---

> ### Author Response · Authors · 2024-11-22
> **Response to Reviewer FU4w**
>
> Thank you for your positive feedback and insightful comment! In regard to the weaknesses and questions raised, we provide our point-to-point responses below:
>
> ---
>
> **Q1:** Question about the decoupling implementation and detailed comparisons with existing works.
>
> **A1:**
> Previous works, such as SSUL, use pseudo-labels and saliency maps to decouple background regions and mitigate the background shift challenge. However, they do not fully address the challenge of semantic shift due to incomplete pseudo labeling and decoupling.
>
> 1. Based on the observation in L233 (L234 in the revised version), an image can be divided into four distinct parts: the region of past classes $\mathcal{C}_{1:t-1}$, target classes $\mathcal{C}\_{t}$, unknown foreground $c'_u$ and pure background $c'_b$. This strategy is adopted by both previous works [1-3] and ours.
> 2. SSUL identifies unknown classes $c'_u$ from the background and employs extra parameters for prediction, which is optimized together with other target class heads. It inevitably introduces noise and affects the learning of target classes, as discussed in L148-154 in our paper.
> 3. **Essentially, we fundamentally analyze the characteristics of noisy semantics and applies a decoupled, divide-and-conquer strategy to learn the noisy semantics**. Specifically, we introduce two separate branches to decouple the learning of these classes:
> - **Temporary branch**: This branch learns the target classes $\mathcal{C}_t$ and non-target foregrounds $c_f$, which change dynamically across each incremental step.
> - **Permanent branch**: This branch learns the pure background $c'_b$ and unseen foreground $c'_u$, which persist throughout the incremental learning process.
>
> We also provide a visualization of semantic decoupling in Figure 7.
>
> ---
>
> **Q2:** Details of memory buffer with image-level samples.
>
> **A2:** We use a shared memory buffer for both the image posterior branch and the segmentation branch. Below, we provide a detailed explanation of how this memory buffer is constructed, updated, and how it storages samples：
>
> 1. **Memory Construction and Update**: Given the memory size $\mathcal{M}$ and the number of already seen classes $|\mathcal{C}\_{1:t}|$, the memory buffer is constructed before step 1 with $\mathcal{M} // |\mathcal{C}_{1:t}|$ samples per class. Once initialized, it is updated before the start of each new step, as done in previous works [1-3]. The update of our memory buffer follows a class-balanced sampling, which is explained in detail in Section 3.5.
> 2. **Data Storage**: For raw data, IPSeg directly stores the image paths in a JSON file, as done in previous works [1-3]. For image-level labels, IPSeg stores the class labels of the images as arrays in the same JSON file with multi-hot encoding, where $1$ indicates the presence of a class and $0$ indicates absence. The memory cost for this is negligible. For pixel-level labels, instead of storing full-class annotations (with data type of *uint8* ) as prior approaches, IPSeg only stores the salient mask, where the background and foreground are labeled as $0$ and $1$, respectively (with data type of *bool* ). Theoretically, the storage space could be reduced to $1/8$.
>
>
>
> ---
>
> **Q3:** The figure presentations are inconsistent. These should be presented by pdf, etc.
>
> **A3:** Thanks for your valuable feedback and careful check regarding the quality of our figure presentations. We replace all blur figures you point out with newer high-resolution PDF version in the the revised manuscript.
>
> ---
>
> **Q4:** Moreover, the authors shall further make some comparisons in terms of training efficacy with the previous method, since they use a memory buff and constitute a two-branches architecture.
>
> **A4:** Please refer to our official comments, we provide a detailed and comprehensive analysis on training and inference cost, thank you.
>
> ---
>
> [1] SSUL: Semantic Segmentation with Unknown Label for Exemplar-based Class-Incremental Learning, NeurIPS 2021.
>
> [2] Mining Unseen Classes via Regional Objectness: A Simple Baseline for Incremental Segmentation, NeurIPS 2022.
>
> [3] CoinSeg: Contrast Inter-and Intra-Class Representations for Incremental Segmentation, ICCV 2023.

---

> > ### Author Response · Authors · 2024-12-02
> > **Response to Reviewer FU4w**
> >
> > Dear Reviewer FU4w.
> >
> > As the author-reviewer discussion period is nearing its end, and since other reviewers have actively engaged in discussions, we would greatly appreciate it if you could review our responses to your comments at your earliest convenience.
> >
> > This will allow us to address any further questions or concerns you may have before the discussion period ends. If our responses satisfactorily address your concerns, please let us know. Thank you very much for your time and effort!
> >
> > Sincerely,
> >
> > The Authors of Submission #731

---

### Official Review · Reviewer_k5js · 2024-11-03

**Soundness:** 4
**Presentation:** 4
**Contribution:** 4
**Rating:** 8
**Confidence:** 5

**Summary:**

This paper addresses the challenges of class incremental semantic segmentation (CISS), where models learn from sequential tasks with changing background semantics, a phenomenon known as semantic drift. The authors identify two key issues—separate optimization and noisy semantics—that significantly hinder CISS performance. To tackle these issues, they propose Image Posterior and Semantics Decoupling for Segmentation (IPSeg), which employs two main mechanisms: (1) Image Posterior Guidance: IPSeg uses image-wise posterior probabilities to guide pixel-wise predictions, mitigating separate optimization issues. (2) Semantics Decoupling: Noisy semantics are split into two groups, stable, static semantics and dynamic, temporary semantics, each handled by separate branches with distinct life cycles. Experiments on the Pascal VOC 2012 and ADE20K datasets show that IPSeg outperforms existing approaches, particularly in challenging long-term scenarios, and demonstrates improved balance between learning plasticity and memory stability.

**Strengths:**

## 1. Strong Motivation

The paper presents a well-defined and convincing motivation for addressing Separate Optimization, a challenge where earlier-trained task heads can produce disproportionately higher scores compared to later-trained heads for visually similar categories. This issue is thoroughly analyzed, with compelling visual support in Figures 1, 5, and 6.

## 2. Comprehensive Analysis

The authors provide a detailed ablation of each proposed component, supported by robust quantitative and qualitative analyses. This thorough exploration effectively demonstrates the contribution of each component to the overall method’s performance.


## 3. State-of-the-Art Performance

The proposed method, IPSeg, demonstrates state-of-the-art performance on the VOC2012 and ADE20K datasets, with results that convincingly surpass existing approaches.

## 4. Clear and Structured Presentation

The paper is well-written and carefully structured. The problem setting and motivation are clearly articulated, and each proposed component is thoroughly explained and ablated, enhancing the paper’s clarity and rigor.

**Weaknesses:**

Most of my concerns raised are addressed within the paper’s appendix, which provides comprehensive additional supportive analysis.

## Recommentation
I conclude that this paper is solid and compelling, meeting the high standards expected for ICLR. My initial recommendation is to Accept. I will finalize the rating after a discussion with the authors and other reviewers.

**Questions:**

N/A

---

> ### Author Response · Authors · 2024-11-22
> **Response to Reviewer k5js**
>
> Thank you for your positive comments and recommendation to our work. We look forward to any further discussions and will make all essential improvements based on your and other reviewers' feedbacks.
>
> Best wishes!

---

> > ### Comment · Reviewer_k5js · 2024-12-02
> >
> > After checking all the reviews and rebuttals, I maintained the original rating. I strongly recommend that the authors improve the final version based on the reviews.

---

### Official Review · Reviewer_wQeR · 2024-11-03

**Soundness:** 2
**Presentation:** 2
**Contribution:** 2
**Rating:** 5
**Confidence:** 5

**Summary:**

The paper proposes IPSeg, an innovative framework for addressing semantic drift in Class-Incremental Semantic Segmentation (CISS). IPSeg leverages two main strategies: (1) Image Posterior (IP) guidance to mitigate errors from independent task optimization, and (2) Permanent-Temporary Semantics Decoupling to handle noisy semantics. These mechanisms allow IPSeg to retain past knowledge while learning new classes, achieving significant improvements in segmentation performance on Pascal VOC and ADE20K benchmarks.

**Strengths:**

1. This approach enhances pixel-level accuracy by leveraging global image-level predictions. Specifically, by introducing the image posterior guidance mechanism, IPSeg aims to mitigate the separate optimization issue in CISS.
2. The decoupling of stable and temporary semantic components is well-conceived and allows the model to handle both static background and dynamic foreground objects across incremental steps.
3. IPSeg outperforms various state-of-the-art methods across different incremental segmentation scenarios.

**Weaknesses:**

1. The issue of separate optimization has been investigated by [2]. In that paper, the scale inconsistency issue "earlier incremental task heads may have larger output scales than the later heads, especially in similar classes" was alleviated by logit manipulation. Therefore, the claim that "separate optimization does not attract any attention" is inaccurate.
2. The introduction of additional components like the image posterior branch and decoupled semantics increases model complexity. This may hinder its applicability for real-time or resource-constrained applications. Efficiency comparisons, such as computational complexity, iterations required for convergence, and finetuning time cost, should be conducted.
3. Salient object detector is used to find out the foreground regions, which will incur additional computational costs. Moreover, it brings additional information and is unfair to other competing methods that do not use it.
4. The memory buffer may raise scalability and privacy concerns, especially when scaling to larger datasets and storing sensitive user data.
5. This paper missed several SOTA methods [1-2] in experimental comparison.

[1] Yang, Ze, et al. "Label-guided knowledge distillation for continual semantic segmentation on 2d images and 3d point clouds." Proceedings of the IEEE/CVF International Conference on Computer Vision. 2023.

[2] Kim, Beomyoung, Joonsang Yu, and Sung Ju Hwang. "ECLIPSE: Efficient Continual Learning in Panoptic Segmentation with Visual Prompt Tuning." Proceedings of the IEEE/CVF Conference on Computer Vision and Pattern Recognition. 2024.

**Questions:**

1. The permanent branch aims to learn dummy labels representing “pure” background and unknown objects. However, the definition of unknown objects are ever-changing. For instance, "cat" can be an unknown class in step 1, and then a target class in step 2, and afterwards become a past seen class in step 3. I don't see the rationale why it is called the permanent branch. Generally, only consistent class definitions across all steps can be regarded as permanent semantics.
2. How are the final prediction maps generated from the permanent and temporary branches? The detail procedures should be elaborated.
3. The previous-class image labels are not available at the current step. How to supervise the image posterior in each step?
4. It seems like temporary branches predict pure background and other foreground regions in addition to the target classes at the current step. Does it contribute to the final prediction or just the target class scores are used to multiply by the corresponding confidence score from the image posterior branch?

**Details Of Ethics Concerns:**

This paper uses a memory buffer to store the training data seen before, which may cause privacy issues when it comes to some privacy-sensitive scenarios, for instance, medical data, personal identity data, etc.

---

> ### Author Response · Authors · 2024-11-22
> **Response to Reviewer wQeR (1/2)**
>
> Thank you for your positive feedback and insightful comments. In regard to the weaknesses and questions raised, we provide our point-to-point responses below.
>
> ---
>
> **W1:** The claim that "separate optimization does not attract any attention" is inaccurate.
>
> **A1:** We sincerely thank the reviewer for pointing out this oversight. We notice that ECLIPSE [1] also focuses on this challenge and proposes the logit manipulation method.  We also revise the corresponding statement. Besides, though focusing on the same challenge, IPSeg differs ECLIPSE from its solution:
> 1. **ECLIPSE** is a prompt-based method built on the Mask2Former network, which encounters an error propagation problem after **freezing the old prompts**. To address this, ECLIPSE incorporates logit manipulation to leverage common knowledge across the classes.
> 2. **IPSeg** is an architecture-based approach built on DeepLabV3 network. We provide a detailed analysis of the issues arising from **freezing the old classification heads**. IPSeg introduces an image posterior branch to explicitly introduce informative image-level class knowledge into segmentation network and directly overcome separate optimization challenge.
>
> ---
> **W2:** Efficiency comparisons, such as computational complexity, iterations required for convergence, and finetuning time cost.
>
> **A2:**  Please refer to our official comment, we provide a detailed and comprehensive analysis on training and inference cost, thanks.
>
> ---
> **W3:** Salient object detector will incur additional computational costs and brings additional information and is unfair to other competing methods that do not use it.
>
> **A3:**
> 1. **Computational costs**：To make saliency maps as auxiliary information, we use an off-the-shelf salient object detector to pre-compute the saliency maps for the entire training set. This process is performed only once, takes less than 10 minutes for Pascal VOC and 30 minutes for ADE20K, and has no impact on inference time.
> 2. **Fair comparison**: We notice SOTA methods [2-4] all introduce additional information to enhance the model's recognition capability. Following these methods, IPSeg utilizes the same saliency maps to ensure a fair comparison.
>
> ---
> **W4:** The memory buffer may raise scalability and privacy concerns, especially when scaling to larger datasets and storing sensitive user data.
>
> **A4:** We appreciate the reviewer's concern regarding memory buffers, scalability, and privacy.
>
> As for the size of memory buffer and scalability, IPSeg sets $\mathcal{M}=100$ for VOC and $\mathcal{M}=300$ for ADE20K, strictly following the same setting as SSUL, MicroSeg and CoinSeg.
>
> In scenarios with privacy constraints or limited data scalability, we also provide the **data-free version of IPSeg** (denoted as "IPSeg w/o M" in Table 1) as an alternative with minor performance loss. For privacy concerns, we discuss the limitations of using the memory buffer and its potential social impact in our "Conclusions" section. We agree with that the privacy issues must be handled with caution in artificial intelligence applications.
>
> ---
> **W5:** This paper missed several SOTA methods in experimental comparison.
>
> **A5:** We provide an experimental comparison on Pascal VOC 2012 and ADE20K below, where IPSeg still achieves competitive performance, particularly in long-term incremental tasks （ADE 100-5 and ADE 100-10）.
>
> | ADE | Backbone | Architecture | 100-5 |||100-10 |||100-50 |||50-50 |||
> | - | - | - |-|-|-|-|-|-|-|-|-|-|-|-|
> | ECLIPSE     | Resnet-101     | Mask2former     |43.3|16.3|34.2|43.4|17.4|34.6|45.0|21.7|37.1|-|-|-|
> |LGKD+PLOP|Resnet-101|Deeplab V3|-|-|-|42.1|22.0|35.4|43.6|25.7|37.5|49.4|29.4|36.0|
> |IPSeg|Resnet-101|Deeplab V3|42.4|22.7|35.9|42.1|22.3|35.6|41.7|25.2|36.3|47.3|32.7|33.6|
> |IPSeg|Swin-B|Deeplab V3|43.2|30.4|38.9|43.0|30.9|39.0|43.8|31.5|39.7|51.1|34.8|40.3|
>
> |VOC|Backbone|Architecture|15-5|||15-1|||10-1|||2-2|||
> |-|-|-|-|-|-|-|-|-|-|-|-|-|-|-|
> | ECLIPSE     | Resnet-101  | Mask2former|-|-|-|-|-|-|-|-|-|-|-|-|
> |LGKD+PLOP|Resnet-101|Deeplab V3|75.2|54.8|71.1|69.3|30.9|61.1|-|-|-|-|-|-|
> |IPSeg|Resnet-101|Deeplab V3|79.5|71.0|77.5|79.6|58.9|74.7|75.9|66.4|71.4|62.4|61.0|61.2|
> |IPSeg|Swin-B|Deeplab V3|83.3|73.3|80.9|83.5|75.1|81.5|80.3|76.7|78.6|73.1|72.3|72.4|
>
> These two methods do not hold comprehensive incremental experiments as IPSeg does. Though they have performance advances in their respective domains, differences in focusing task scenarios may limit their direct comparability with IPSeg in the context of our study. Upon this point, it is unfair for them to compare with ours. We also add them into the main results in Table 1 and Table 2 for comprehensive comparisons as your suggestions.

---

> > ### Comment · Reviewer_wQeR · 2024-11-23
> > **Request for Data-Free IPSeg w/o M Results on ADE20k**
> >
> > Please do not list the results **without denoting whether memory is used or not** in the same table in A5. It is not fair to do so. Additionally, I would like to request for **Data-Free IPSeg w/o M** results on ADE20k.

---

> ### Author Response · Authors · 2024-11-22
> **Response to Reviewer wQeR (2/2)**
>
> **Q1:** I don't see the rationale why it is called the permanent branch. Generally, only consistent class definitions across all steps can be regarded as permanent semantics.
>
> **A1:** In our work, the terms "permanent" and "temporary" do not refer to specific classes but depend on their learning cycle throughout the incremental stages. For the concepts existing across all incremental steps, we use the term "permanent" to describe them and the "permanent branch" to indicate the branch consistently learning them across the whole incremental steps.
>
> There are three components needed to be identified once a new incremental step begins: target classes set $\mathcal{C}\_{t}$, the pure background $c_b'$, and the unknown set $c_u'$. Across all incremental steps, $\mathcal{C}_{t}$ changes drastically, $c_b'$ keeps fixed and $c_u'$ shrinks but will not disappear. Compared with the ever changing target class set $\mathcal{C}_t$, the pure background $c_b'$ and the unknown set $c_u'$ exist across all incremental steps.
>
> Further, based on your example, what happens in our IPSeg is as follows: "cat" is included in the unknown set in step 1 to be learned by the permenant branch. In step 2, "cat" is removed from the unknown set and transformed into target class set. After step 2, "cat" is only regarded as a seen class. This transformation process happens on all target classes across all incremental steps.
>
>
> ---
>
> **Q2:** How are the final prediction maps generated from the permanent and temporary branches?
>
> **A2:** As shown by the green lines in Figure 2, the permenant branch $\phi_p$ outputs prediction for the background $c_b'$, and the temporary branch $\phi_i$ (i=1,2,...,t) outputs predictions for the target classes $\mathcal{C}\_t$. And the pixel-level prediction $\phi_{0:T}(h_\theta(x_i))$ can be writen as:
> $$ \phi_{0:T}(h_\theta(x_i))= Con(   \phi_{p,bg}(h_\theta(x_i)) ,   \phi_{1:T,target}(h_\theta(x_i))   ).$$ Where $Con$ is the concatenate operation, $\phi_{p,bg}(h_\theta(x_i))$ and $\phi_{1:T,target}(h_\theta(x_i))$ represent the background prediction from the permenant branch and the target classes prediction from the temporary branch. And this pixel-level prediction is then producted by image posterior probability to form the final predition maps as Eq.3 and Eq.4.
>
> In the revised manuscript, we provide a detailed description of the prediction map generation process.
>
> ---
>
> **Q3:** The previous-class image labels are not available at the current step. How to supervise the image posterior in each step?
>
> **A3:** As mentioned in L204-207 (L206-210 in the revised version), image posterior branch uses mixed data from memory buffer and current training dataset of each step. The supervision is derived from the knowledge of all seen classes in the mixed data, which mainly comes from:
> 1. Samples from memory buffer are saved with their image-level labels of corresponding stages, contributing to knowledge of previous classes.
> 2. IPSeg uses image-level pseudo-label of current data to capture the knowledge of old classes.
> 3. The ground truth of current task data on target classes is available.
>
> To investigate how these forms of supervision improve performance, we provide detailed ablation in Table 4.
>
>
> ---
>
> **Q4:** Does the temporary branches predictions contribute to the final prediction or just the target class scores are used to multiply by the corresponding confidence score from the image posterior branch?
>
> **A4:**
> Yes, there are two roles the temporary branch plays in contributing to the final prediction. First is the target class predictions $\phi_{1:T,target}(h_\theta(x_i))$, and it is used as our answer to Q2 above. Second is the prediction of other foreground regions $c_f$, and IPSeg utilizes $c_f$ from temporary branches of each step to filter out erroneous outputs during inference as shown in L265–271 (L277-285 in the revised version). While the pure background prediction within each temporary branch does not contibute to the final predictions, and it only helps the model distinguish the target classes during training.
>
> ---
>
> [1] ECLIPSE: Efficient Continual Learning in Panoptic Segmentation with Visual Prompt Tuning, CVPR 2024.
>
> [2] SSUL: Semantic Segmentation with Unknown Label for Exemplar-based Class-Incremental Learning, NeurIPS 2021.
>
> [3] Mining Unseen Classes via Regional Objectness: A Simple Baseline for Incremental Segmentation, NeurIPS 2022.
>
> [4] CoinSeg: Contrast Inter-and Intra-Class Representations for Incremental Segmentation, ICCV 2023.
>
> [5] Label-guided knowledge distillation for continual semantic segmentation on 2d images and 3d point cloud, ICCV 2023.

---

> ### Author Response · Authors · 2024-11-28
> **Response to Reviewer wQeR**
>
> **Q:** Please do not list the results without denoting whether memory is used or not in the same table in A5. It is not fair to do so. Additionally, I would like to request for Data-Free IPSeg w/o M results on ADE20k.
>
> **A:** Thanks for your tips. Here, we provide the results denoting whether memory is used or not below. Besides, we reorganize Table 2 and add the results of "**IPSeg w/o M**" for comprehensive and fair comparison.
>
>
>
> | ADE20K | memory|Backbone | Architecture | 100-5 |||100-10 |||100-50 |||50-50 |||
> | -------- | :-:|------- | -------- |-|-|-|-|-|-|-|-|-|-|-|-|
> |||||0-100|101-150|all|0-100|101-150|all|0-100|101-150|all|0-50|51-150|all|
> | ECLIPSE |×    | Resnet-101     | Mask2former     |43.3|16.3|34.2|43.4|17.4|34.6|45.0|21.7|37.1|-|-|-|
> |LGKD+PLOP|×|Resnet-101|Deeplab V3|-|-|-|42.1|22.0|**35.4**|43.6|25.7|**37.5**|49.4|29.4|**36.0**|
> |SSUL|×|Resnet-101|Deeplab V3|39.9|17.4|32.5|40.2|18.8|33.1|41.3|18.0|33.6|48.4|20.2|29.6|
> |MicroSeg|×|Resnet-101|Deeplab V3|40.4|20.5|33.8|41.5|21.6|34.9|40.2|18.8|33.1|48.6|24.8|32.9|
> |PLOP+NeST|×|Resnet-101|Deeplab V3| 39.3|17.4|32.0|40.9|22.0|34.7|42.4|24.3|36.3|48.7|27.7|34.8|
> |IPSeg w/o M |×|Resnet-101|Deeplab V3|41.0|22.4|**34.8**|41.0|23.6|35.3|41.3|24.0|35.5|46.7|26.2|33.1|
> |
> |IPSeg|√|Resnet-101|Deeplab V3|42.4|22.7|35.9|42.1|22.3|35.6|41.7|25.2|36.3|47.3|26.7|33.6|
>
>
> |ADE20K  |memory| Backbone | Architecture | 100-5 |||100-10 |||100-50 |||50-50 |||
> | -------- | :-:| -------- | -------- |-|-|-|-|-|-|-|-|-|-|-|-|
> |||||0-100|101-150|all|0-100|101-150|all|0-100|101-150|all|0-50|51-150|all|
> |SSUL|×|Swin-B|Deeplab V3|41.3|16.0|32.9|40.7|19.0|33.5|41.9|20.1|34.6|49.5|21.3|30.7|
> |CoinSeg|×|Swin-B|Deeplab V3|43.1|24.1|36.8|42.1|24.5|36.2|41.6|26.7|36.6|49.0|28.9|35.6|
> |PLOP+NeST|×|Swin-B|Deeplab V3|39.7|18.3|32.6|41.7|24.2|35.9|43.5|26.5|37.9|50.6|28.9|36.2|
> |IPSeg w/o M|×|Swin-B|Deeplab V3|43.1|26.2|**37.6**|42.5|27.8|**37.6**|43.2|29.0|**38.4**|49.3|33.0|**38.5**|
> |
> |IPSeg|√|Swin-B|Deeplab V3|43.2|30.4|38.9|43.0|30.9|39.0|43.8|31.5|39.7|51.1|34.8|40.3|
>
> From the results, we can draw two key conclusions:
>
> 1. **Comparable Performance Without Memory Buffer**: Even without a memory buffer, our data-free version demonstrates close performance to our data-replay version, exhibiting only a minor performance loss (up to 1.1% when using ResNet-101 on ADE20K). Our method maintains robust and consistent performance across all settings, whether using memory buffer or not.
>
> 2. **Competitive Performance in Long-Term Incremental Scenarios**: The data-free version of IPseg (IPseg w/o M) achieves competitive performance with both ResNet-101 and Swin-B backbones, especially in challenging long-term incremental scenarios (e.g., ADE 100-5, ADE 100-10).

---

> ### Author Response · Authors · 2024-12-02
> **Response to Reviewer wQeR**
>
> Dear Reviewer wQeR.
>
> As the author-reviewer discussion period is nearing its end, and since other reviewers have actively engaged in discussions, we would greatly appreciate it if you could review our responses to your comments at your earliest convenience.
>
> This will allow us to address any further questions or concerns you may have before the discussion period ends. If our responses satisfactorily address your concerns, please let us know. Thank you very much for your time and effort!
>
> Sincerely,
>
> The Authors of Submission #731

---

> ### Comment · Reviewer_wQeR · 2024-12-03
> **Concerns with the Key Contribution**
>
> I would like to thank the authors for their efforts in providing additional experiments on ADE20K. However, the results bring me some concerns about the key contribution "the image-level classification (IP) suffers less forgetting than the segmentation".
>
> ## Concerns:
>
> - **IPSeg w/o M** performs **inferior** compared to **LGKD+PLOP** on the three standard settings **ADE20K 100-10**, **100-50** and **50-50**, when no memory is used. Could the author explain the underlying reasons? As mentioned by **Reviewer zxHa**, **the image-level classification (IP) suffers less forgetting than the segmentation** is regarded as the **key contribution** of this paper. If this is the case, I believe **IPSeg w/o M** should definitely outperform **LGKD+PLOP** [5] on ADE20K **without any memory**. This is my main concern, and I hope the authors can provide convincing response.
>
>
> | Model            | Memory | Backbone   | Architecture | |100-10 || |100-50| | |50-50 ||
> |------------------|:-:|:-:|:-:|:-:|:-:|:-:|:-:|:-:|:-:|:-:|:-:|:-:|
> | | | | | *0-100*  | *101-150* | *all*    | *0-100*  | *101-150* | *all*    | *0-100*  | *101-150* | *all*    | *0-50*   | *51-150* | *all*    |
> | LGKD+PLOP [5]        | ×      | Resnet-101 | Deeplab V3  | **42.1**   | 22.0    | **35.4**   | **43.6**  | **25.7**    | **37.5**   | **49.4**   | **29.4**   | **36.0**   |
> | IPSeg w/o M      | ×      | Resnet-101 | Deeplab V3   | 41.0   | **23.6**    | 35.3   | 41.3   | 24.0    | 35.5   | 46.7   | 26.2    | 33.1   |
>
> ---
> - The computational cost is considerable compared to SSUL (**137.1G vs. 94.9G**), indicating a **44.5\% increase**. I am not convinced by the cost-efficiency.
>
> ---
> ## Questions:
>
> - From my perspective, ECLIPSE does not suffer from error propagation (L157). Could the reviewer comment on this?
> - The subscript $\phi_{1: t-1}\left(h_\theta\left(x_i^{m, t}\right)\right)$ used for $\tilde{\mathcal{Y}}$ is inconsistent to the subscript $i$ used for $\mathcal{Y}^t$. I suppose i to be the i-th input image. However, $\phi_{1: t-1}\left(h_\theta\left(x_i^{m, t}\right)\right)$ is related to the class index predicted by previous heads.
>
> ---
> [5] Yang, Ze, et al. "Label-guided knowledge distillation for continual semantic segmentation on 2d images and 3d point clouds." Proceedings of the IEEE/CVF International Conference on Computer Vision. 2023.

---

> > ### Author Response · Authors · 2024-12-04
> > **Further Response to wQeR (1/2)**
> >
> > Dear reviewer,
> >
> > Thanks for your timely response and your interests on our work. To your concerns and questions, here we explain and answer them one by one.
> >
> > ---
> >
> > **C1:** Could the author explain the underlying reasons? As mentioned by Reviewer zxHa, the image-level classification (IP) suffers less forgetting than the segmentation is regarded as the key contribution of this paper. If this is the case, I believe IPSeg w/o M should definitely outperform LGKD+PLOP on ADE20K without any memory. This is my main concern, and I hope the authors can provide convincing response.
> >
> >
> > **A1:**
> > It is not entirely fair to compare our work directly with LGKD without considering their respective focuses and properties. These two methods address different challenges in CISS. Specifically, our method is optimized by leveraging a memory buffer to address the separate optimization challenge. The key features of our work are outlined below:
> >
> > Firstly, we emphasize the data-replay version of IPSeg as the main experimental result in our paper (Table 1 and Table 2) because **the Image Posterior (IP) Branch is originally designed to operate with a memory buffer**:
> > - The IP branch is supervised using  $\mathcal{Y}^{t}\_i \cup Y\{(\phi_{1:t-1}(h_\theta(x^{m,t}\_i)))\}$  , where the latter term $Y\{(\phi_{1:t-1}(h_\theta(x^{m,t}_i)))\}$ is derived from pixel predictions generated by segmentation heads. The accuracy of these segmentation heads relies significantly on the memory buffer. This design is ablated in Table 4.
> > - Additionally, the samples stored in the memory buffer provide accurate image-level supervision for past classes, enabling the IP branch to suffer less forgetting.
> >
> > Secondly, **removing the memory buffer leads to degradation.** Without the memory buffer, the IP branch experiences degradation in image classification, which, combined with the degradation of the segmentation branch, leads to an overall performance decline in IPSeg w/o M.
> >
> > **In summary, IPSeg is fundamentally designed with memory buffer, which has a critical contribution to our work in effectively mitigating forgetting. And the data-free version of IPSeg (IPSeg w/o M) is not the core contribution of our work but rather a supplementary component.** Instead, we present it as an alternative option for scenarios requiring privacy protection.
> >
> > ---
> >
> > **C2:** The computational cost is considerable compared to SSUL (137.1G vs. 94.9G), indicating a 44.5% increase. I am not convinced by the cost-efficiency.
> >
> > **A2:**
> > Our work truly introduces higher computational cost compared to our baseline method. **But the cost-efficiency also needs to be considered in comparison with the sota method, where IPSeg offers significant advantages in training cost while maintaining comparable inference speed (FPS) and compute cost (FLOPs)**. This can be found in our cost-efficiency results in official comment.
> >
> > **Compared with the baseline, our additional branch increases the model's FLOPs, but it has minimal impact on inference speed (FPS).** The increase in FLOPs mainly stems from IPSeg’s use of image-level predictions to guide final outputs. Specifically, IPSeg broadcasts image-level predictions to match the shape of pixel-level logits and combines them through element-wise multiplication. Although this introduces dense computational operations, these are inherently parallelizable and can be extensively optimized and accelerated by GPUs, ensuring that the inference speed remains largely unaffected. **Besides, relying solely on a single metric to evaluate cost is neither comprehensive nor objective.** Inference speed (FPS) is a crucial factor that must also be considered, as it serves as the most direct measure of real-time ability.
> >
> > **Moreover, it can not be ignored that IPSeg achieves substantial performance improvements with only a modest increase in inference costs compared to the baseline.** According to the reported results, IPSeg operates at an FPS of 27.3, which is approximately 6 FPS lower than SSUL-M's 33.7, and requires 6.2G of GPU memory compared to SSUL-M’s 5.3G. However, IPSeg delivers a remarkable overall mIoU of **81.5**, significantly surpassing SSUL-M’s 71.9, representing a **9.6 improvement in performance**.

---

> > > ### Author Response · Authors · 2024-12-04
> > > **Further Response to wQeR (2/2)**
> > >
> > > **Q1:** From my perspective, ECLIPSE does not suffer from error propagation (L157). Could the reviewer comment on this?
> > >
> > > **A1:**
> > > Thanks, we also agree that ECLIPSE address the error propogation problem by its proposed logit manipulation, as we wrote in our earlier response,
> > > > ECLIPSE is a prompt-based method built on the Mask2Former network, which encounters an error propagation problem after freezing the old prompts. To address this, ECLIPSE incorporates logit manipulation to leverage common knowledge across the classes.
> > >
> > >
> > > In L157, we claimed that **ECLIPSE points out and researchs the error propagation problem** instead of suffering from error propagation. Based on your feedback, we will revise our description as below:
> > > > Previous work points out the challenge that freezing parameters from the old stage can preserve the model's prior knowledge but will lead to error propagation and confusion between similar classes, and proposes logit manipulation to solve this challenge.
> > >
> > >
> > > We hope this newly refined version can clearly express our attitude and views without causing any confusion or ambiguity.
> > >
> > > ---
> > >
> > > **Q2:** The subscript $\phi_{1:t-1}(h_\theta(x_i^{m,t}))$ used for $\tilde{\mathcal{Y}}$ is inconsistent to the subscript $i$ used for $\mathcal{Y}^t$. I suppose $i$ to be the i-th input image. However, $\phi_{1:t-1}(h_\theta(x_i^{m,t}))$ is related to the class index predicted by previous heads.
> > >
> > > **A2:**
> > > Thanks for your timely reminder of inconsistent subscript.
> > > This notation is used in our equation.2 with its explanation in L211-213:
> > > $$\mathcal{L}\_{\text{ IP}} = \mathcal{L}\_{\text{ BCE}}(\hat{\mathcal{Y}}^{t}\_i, \tilde{\mathcal{Y}}^{t}\_i) = \mathcal{L}\_{\text{ BCE}}(\psi(h_\theta(x^{m,t}\_i)), \tilde{\mathcal{Y}}^{t}\_i), ~~\tilde{\mathcal{Y}}^{t}\_i = \mathcal{Y}^{t}\_i \cup \tilde{\mathcal{Y}}\_{\phi_{1:t-1}(h_\theta(x^{m,t}\_i))}$$
> > > > ... and pseudo label $\tilde{\mathcal{Y}}\_{\phi_{1:t-1}(h_\theta(x^{m,t}\_i))}$ on past seen classes $\mathcal{C}\_{1:t-1}$.
> > >
> > > We use $\tilde{\mathcal{Y}}\_{\phi_{1:t-1}(h_\theta(x^{m,t}\_i))}$ to represent the image-level pseudo labels on past classes that are derived from previous segmentation heads.
> > >
> > > We agree that it is not consistent with already used subscript $i$ for image-index as you mentioned. Considering this inconsistency, we decide to revise this notation into $\{\phi_{1:t-1}(h_\theta(x^{m,t}_i))\}$ with using set operator $\{\cdot\}$ on class channel instead. Correspondingly, we also revise the equation.2 and its explanation into the following form:
> > >
> > > $$\mathcal{L}\_{\text{ IP}} = \mathcal{L}\_{\text{ BCE}}(\hat{\mathcal{Y}}^{t}\_i, \tilde{\mathcal{Y}}^{t}\_i) = \mathcal{L}\_{\text{ BCE}}(\psi(h_\theta(x^{m,t}\_i)), \tilde{\mathcal{Y}}^{t}\_i), ~~\tilde{\mathcal{Y}}^{t}\_i = \mathcal{Y}^{t}\_i \cup Y{(\phi_{1:t-1}(h_\theta(x^{m,t}\_i)))\}$$
> > > > ... and pseudo label $Y(\{\phi_{1:t-1}(h_\theta(x^{m,t}_i)))\}$ on past seen classes $\mathcal{C}\_{1:t-1}$, $Y( \cdot\)$ is set operator on class channel.
> > >
> > > Thanks again for your careful and rigorous check, this indeed further helps us to improve the quality of our manuscript.

---

### Official Review · Reviewer_zxHa · 2024-11-04

**Soundness:** 3
**Presentation:** 3
**Contribution:** 3
**Rating:** 8
**Confidence:** 4

**Summary:**

This paper indicates two key issues within semantic drift, separate optimization, and noisy semantics, for class-incremental segmentation. The authors propose a method called IPSeg, including image posterior probabilities and semantics decoupling. This paper conducts extensive experiments on replay-based and non-replay-based scenarios and shows significant improvement under extremely long incremental learning steps.

**Strengths:**

This paper identifies two issues in the semantic drift for class-incremental segmentation and proposes two components for each problem, respectively. Extensive experiments demonstrate the effectiveness of the proposed method.

**Weaknesses:**

The writing and clarification could be simplified in Section 3.3 and Section 3.4, especially for Equation 5. In Figure 2, the targets of the permanent and temporal branches are not clear.

**Questions:**

- This paper uses image posterior probability to rectify the class prediction. However, the subnetwork for image-level classification is trainable during different steps. I wonder why the classification subnetwork does not suffer from catastrophic forgetting. Is there any evaluation of the performance of the classification subnetwork during multiple steps?
- In Figure 2, the feature extractor is frozen.  What is the pre-trained knowledge of the backbone from, the first training step or the ImageNet?
- The mechanism of image posterior probability is similar to the way used in transformer-based mask prediction, like Incrementer [1]. Is there any comparison and discussion with it?
- Because this work modifies the inference procedure of the mask, please report the training and inference costs, such as training time and memory.
- This method uses salient maps derived from an off-the-shelf model as one of the targets. The salient maps are suitable and useful for the VOC dataset because most objects can be detected by the salient model. However, for the more challenged dataset, ADE20K, many semantic regions can not be identified by the salient model, such as grassland in the supplementary material Figure 9. Please study the effect of the salient maps supervision and the quality of the salient maps on the ADE20K dataset. A suggestion: you may replace the salient maps with the results of the SAM model [2].
- In Table 2, why is the replay-based IPSeg (33.6) worse than data-free-based PLOP+NeST (34.8)? Is there any insight or explanation?

[1] Incrementer: Transformer for class-incremental semantic segmentation with knowledge distillation focusing on old class, CVPR 2023
[2] Segment Anything, ICCV 2023

---

> ### Author Response · Authors · 2024-11-22
> **Response to Reviewer zxHa (1/2)**
>
> Thank you for your positive feedback and constructive comments. In regard to the weaknesses and questions raised, we provide our point-to-point responses below:
>
> ---
> **Q1:** Why the classification subnetwork does not suffer from catastrophic forgetting? Is there any evaluation of the performance of the classification subnetwork?
>
> **A1:**
> **Evaluation on classification subnetwork**: We provide an evaluation of the performance of the classification subnetwork after all steps as below. We hold three experiment settings: "**Seq**" refers to training the subnetwork sequentially without any additional tricks, indicating the worst case suffering from catastrophic forgetting, "**Ours**" refers to the training setting used in our paper, and "**Joint**" refers to training with all task data jointly as the upper bound.
> |       | Precision| Recall   | F1 |
> | :--------: |:--------: | :--------: |:---: |
> | Seq |   55.62%   | 24.67%     | 23.78% |
> | Ours | 78.28%     | 87.03%     | 80.68% |
> | Joint |  89.96%    | 90.00%     | 89.89% |
>
> **Analysis**: It is obvious that the classification subnetwork suffers little forgetting. We achieve this performance mainly owing to two specific settings:
> 1. **Mixed training data**: As mentioned in L204-207 (L206-209 in the revised version), the classification subnetwork uses data from both the memory buffer and the current task dataset for training. Compared to fine-grained task heads, it is easier for the subnetwork to learn image-level knowledge from memory buffer.
> 2. **Image-level pseudo-label for supervision**:  As claimed in L210-213 (L212-214 in the revised version), IPSeg introduces image-level pseudo-label on past classes as supervision to mitigate catastrophic forgetting challenge. The ablation result in Table 4 partially reflects the effectiveness of this design.
>
> ---
> **Q2:** What is the pre-trained knowledge of the backbone from, the first training step or the ImageNet?
>
> **A2:** Both. We use the ImageNet pretrained Resnet-101 and Swin-B to intilize our backbone. Then, we finetune the backbone at the first step and freeze it in the subsequent steps following previous works [1-2].
>
> ---
> **Q3:** The differences between the mechanism of image posterior probability and transformer-based mask prediction, like Incrementer.
>
> **A3:** Thanks for your reminder. We list the commons and differences between Incrementer and IPSeg as follows:
>
> Commons:
>
> 1. Mask generation: The processes of mask generation in Incrementor and IPSeg are similar. They all generate mask prediction by fusing dense visual information with sparse class information.
> 2. Class information: The class tokens in Incrementer and image posterior probabilities in IPSeg both contain classes information.
>
>
> Differences:
> 1. Inplementation details: Incrementer computes cosine similarity between visual embedding and class embedding to generate mask prediction. While IPSeg multiplies dense pixel-level class predictions with image-level class predictions to generate the final outputs.
> 2. Mechanism: In Incrementer, class embeddings are responsible for class information while visual embeddings are class agnostic. But in IPSeg, both the pixel-level and image-level predictions are  class-related, and the latter plays a role in introducing image-level class guidance to rectify the former.

---

> > ### Comment · Reviewer_zxHa · 2024-11-23
> >
> > Thanks for the authors' response.
> > Regarding the first point, "Compared to fine-grained task heads, it is easier for the subnetwork to learn image-level knowledge from memory buffer," I wonder how the authors reached this conclusion. Would you please report the classification accuracy of two tasks (image-level classification and pixel-level classification)? I understand that the pixel-level classification may have lower accuracy if you count pixel-level accuracy. So, please calculate the image-level accuracy for both tasks. For the segmentation head, if a pixel belongs to class $C$, then the image belongs to class $C$. I hope you demonstrate that the image-level classification can suffer less forgetting than the segmentation, which is my core concern. Otherwise, I can not understand why the image-level classification can alleviate forgetting in segmentation.

---

> ### Author Response · Authors · 2024-11-22
> **Response to Reviewer zxHa (2/2)**
>
> **Q4:** The training and inference costs, such as training time and memory.
>
> **A4:** Please refer to our official comment, we provide a detailed and comprehensive analysis on training and inference cost, thanks.
>
> ---
>
> **Q5:** The effect of the salient maps supervision and the quality of the salient maps on the ADE20K dataset.
>
> **A5:** It indeed poses a challenge when using saliency maps on ADE20K to identify target regions, specifically for objects of small scale and at the edges of images. According to your concerns, we conduct the ablation study on the ADE20K 100-10 task with the following settings: "**w/o Sal**" uses no saliency map supervision, "**w/ Sal**" uses saliency map supervision as we do in paper, and "**w/ SAM**" uses saliency maps extracted by SAM [3] model. We use the official Grounded SAM [4] code with all class names in ADE20K as text prompts to extract corresponding masks in implementation. Additionally, we also report performance differences on "thing" and "stuff" classes defined in ADE20K panoptic segmentation[5] to investigate the bias of saliency maps on different semantic regions.
>
> | Method    | 0-100    | 101-150  | All  | Things |Stuff |
> | :--------:  | :-:      | :-:      | :-:  |:-:     |:-:   |
> | w/o Sal    |   42.1   |  28.0    | 37.4 |37.2    |37.5  |
> | w/ Sal     |   43.0   |  30.9    | 39.0 |39.1    |37.6  |
> | w/ SAM     |   43.6   |  31.8    | 39.7 |39.7    |38.7  |
>
> Basically, we find the following conclusions:
> 1. Using the default saliency map supervision to implement knowledge decoupling strategy, IPSeg gets a performance improvement by $+1.6$% mIoU. And using saliency maps extracted by SAM further improves IPSeg performance by $+0.7$% mIoU.
> 2. The default setting performs well in identifying "Things" classes but struggles with "Stuff" classes, resulting in a performance gain of $+1.9$% on "Things" classes but merely $+0.1$% on "Stuff" classes. Furthermore, the SAM-based saliency maps provide better supervision for both "Things" and "Stuff" classes, with improvements of $+0.6$% on "Things" and $+1.1$% on "Stuff" compared to "w/ Sal".
>
> In summary, the analysis result confirms that the quality of saliency maps affects model performance on the ADE20K dataset. We will report this conclusion in our supp to inspire future works.
>
> In our paper, IPSeg uses the same saliency map extraction method without more information leakage to ensure a fair comparison with baseline methods. Despite this, IPSeg still achieves SOTA performance compared to SOTA methods like MicroSeg and CoinSeg, which utilize region proposals from more advanced models like Mask2Former as auxiliary information. Thanks for your interesting suggestion.
>
>
>
> ---
> **Q6:** Why is the replay-based IPSeg (33.6) worse than data-free-based PLOP+NeST (34.8)? Is there any insight or explanation?
>
> **A6:** We find the PLOP+NeST outperforms IPSeg in the ADE20K 50-50 setting mainly because of two reasons:
>
> 1. **Unfair Training Epochs**: Compared with IPSeg, NeST [6] requires additional **15** warm-up epochs to initilize new classifiers for each step. These extra training epochs allow NeST to better adapt to new classes, leading to improved performance.
> 2. **Characteristics of the 50-50 Task**: NeST aligns new classifiers with the backbone and adapts to the new class data with extra **15** warm-up epochs, which relies on sufficient training data. In the "50-50" or "100-50" settings, there are a large amount of new classes and training data to help NeST better warm up. However, in long-term challenging tasks such as "100-5" and "100-10", there is not enough new task data to achieve similar warm-up effect, and its performance is not ideal as in "50-50" setting.
>
> Generally speaking, the performance advance on the ADE20K 50-50 setting is highly related to the inconsist training setting in NeST.
>
> ---
>
> [1] SSUL: Semantic Segmentation with Unknown Label for Exemplar-based Class-Incremental Learning, NeurIPS 2021.
>
> [2] Mining Unseen Classes via Regional Objectness: A Simple Baseline for Incremental Segmentation, NeurIPS 2022.
>
> [3] Segment Anything, ICCV 2023.
>
> [4] Grounded SAM: Assembling Open-World Models for Diverse Visual Tasks, Arxiv 2024.
>
> [5] Semantic Understanding of Scenes Through the ADE20K Dataset, IJCV 2019.
>
> [6] Early Preparation Pays Off: New Classifier Pre-tuning for Class Incremental Semantic Segmentation, ECCV 2024.

---

> > ### Comment · Reviewer_zxHa · 2024-11-23
> >
> > Regarding Q5, Grounding SAM can not be used, because it has data leakage when using all class names as prompts. This is only my reminder, not my concern.
> > For Q6, the authors' response is not so convincing. I suggest that the authors retrain the proposed IPSeg in the 50-50 setting using any preferred training schedule to show that the performance can be further improved.

---

> ### Author Response · Authors · 2024-11-28
> **Response to Reviewer zxHa**
>
> **Q1:** please calculate the image-level accuracy for both tasks.
>
> **A1:**
> Based on your concerns, we hold two experiments to answer your questions.
>
> 1. First, we evaluate the image level accuracy performance of the base 15 classes using the image posterior (IP) branch and the segmentation (Pixel) branch at each step on Pascal VOC 15-1 to further investigate their **forggetting** on base classes. "IP" refers to only using the IP branch, "Pixel" refers to only using the segmentation branch, where the class $\mathcal{C}$ exists if a pixel is predicted as $\mathcal{C}$. "Pixel+IP" denotes using them both as our paper does.
>
> | Base Classes ACC (%)    |step 0| step 1| step 2   | step 3 |step 4|step 5|
> | -------- | -------- | -------- |--- |-|-|-|
> | IP | 87.44|86.41|86.99|86.82|86.86|86.29|
> |Pixel|88.17|86.42|86.30|85.43|84.84|84.70|
> |Pixel+IP|93.07|92.24|92.41|91.93|91.95|91.02|
>
> The ablation shows that: **the image-level classification (IP) suffers less forgetting than the segmentation (Pixel)**, and our method (Pixel+IP) shows similar property against forgetting with the help of IP.
>
> 2. Additionally, we also evaluate the image-level accuracy on all seen classes at each step to analyze their performance on both keeping old knowledge and learning new knowledge.
>
> | Seen Classes ACC (%)    |step 0| step 1| step 2   | step 3 |step 4|step 5 (Final)|
> | -------- | -------- | -------- |--- |-|-|:-:|
> | IP | 87.44|82.54|81.14|81.32|82.09|**82.34**|
> |Pixel|88.17|83.56|82.29|78.23|77.60|**76.57**|
> |Pixel+IP|93.07|90.05|90.13|87.30|87.68|**88.03**|
>
> For the segmentation branch, the image-level accuracy of it on all seen classes gradually degrades after learning new classes, performing worse than its accuracy on base classes. This indicates the segmentation branch performs poorly on new classes, which is consistent with our description about separate optimization (L160-161 and L180-182 in the revised manuscript). In contrast, the IP branch experiences less deterioration from separate optimization and help our method maintain a good balance between retaining old knowledge and learning new knowledge. Therefore, the ablation evidently shows that **the IP branch learns image-level knowledge better than the segmentation branch**.
>
> In summary, these experiments demonstrate that, on one hand, **the image classification (IP) branch exhibits higher accuracy** after all steps and **suffers less forgetting**. On the other hand, **the IP branch mainly helps our method mitigate the sparate optimization, effectively improving overall performance**.
>
> Please let us know whether our response solve your questions and concerns.
>
> ---
>
> **Q2:**  For Q6, the authors' response is not so convincing. I suggest that the authors retrain the proposed IPSeg in the 50-50 setting using any preferred training schedule to show that the performance can be further improved.
>
> **A2:** Thanks for your suggestion. We run an experiment on ADE20K 50-50 setting with a longer schedule as same as PLOP+NeST uses and report the result as below. Using the same training schedule, our method shows slight advantage over NeST.
>
> ADE20K 50_50|0-50|51-150|all|
> |-|-|:-:|-|
> PLOP+NeST(75 epochs)|48.7|27.7|34.8|
> IPSeg(60 epochs)|47.3|26.7|33.6|
> IPSeg(75 epochs)|47.7|28.7|35.1|
>
> ---
>
> **Q3:** Regarding Q5, Grounding SAM can not be used, because it has data leakage when using all class names as prompts. This is only my reminder, not my concern.
>
> **A3:**  The SAM model segments all regions of a given image indiscriminately, making it difficult to distinguish between foreground and background areas. Consequently, this poses a challenge for using SAM as salient maps within our methods. To address this issue, we employ the Grounded SAM model instead but introduce information leakage problem as you mentioned. We will further explore related techniques. Thanks for your kind reminder.

---

> ### Comment · Reviewer_zxHa · 2024-11-28
>
> Thanks for the authors' response.
> The authors have clearly demonstrated that image-level classification suffers less forgetting than semantic segmentation, which verifies the motivation of IPSeg. Additionally, the authors show that IPSeg can perform better than PLOP + NeST using the same training epochs. Hope these results can appear in the revision and I decide to raise my rate from 6 to 8.

---

> ### Author Response · Authors · 2024-11-29
> **We greatly appreciate your constructive suggestions for improving our work**
>
> Thank you for your positive feedback on our work and for increasing the score. We greatly appreciate your constructive suggestions for improving our work.
>
> Best wishes.

---

### Author Response · Authors · 2024-11-22
**To all Reviewers**

We thank all reviewers for their great efforts and constructive comments, which help us further improve our work. This official comment contains two part: 1. The analysis on training and inference cost. 2. The summary of our manuscript modification.

---

## 1. Analysis on training and inference cost
Here, we provide a comprehensive analysis of the model parameters, training, and inference costs. We test and report the results of IPSeg, SSUL-M and CoinSeg-M with Swin-B on the VOC 15-1 setting. We set *image_size=512x512, epochs=50, and batch_size=16* in training and *image_size=512x512* for inference. All results are run on RTX 3090 GPU.

1. **Model Parameters**: Using the thop tool, we analyze and compare the trainable parameters for these methods. The increase in parameter sizes is similar across them, with an average of $3.84M$ additional parameters per step. Additionaly, IPSeg has $29.72M$ parameters more than SSUL-M due to the additional image posterior branch.

| Step | 0 | 1 | 2 | 3 | 4 | 5 |
| :-------- | :--------: | :--------: | :--: | :--: | :--: | :--: |
| IPSeg     | 135.92 M    | 139.76 M   | 143.60 M | 147.66 M| 151.28 M| 155.12 M|
| SSUL-M     |  106.20 M   |110.03 M     | 113.89 M|117.95 M | 121.56 M| 125.40 M |
| CoinSeg-M | 107.02 M | 111.15 M | 115.29 M | 119.42 M | 123.55 M | 127.68 M |


2. **Training cost**: The training time and GPU Memory usage of these methods are shown in the table below. Due to the introduced image posterior branch, IPSeg needs more training cost compared with SSUL-M but less than CoinSeg-M.

| Method | Time | GPU usage |
| :-------- | :--------: | :--------:|
| IPSeg     | 9h 14min     | 21.1G     |
| SSUL-M     | 7h 13min    | 19.4G     |
| CoinSeg-M | >15h         |21.3G      |

3. **Inference**: The inference speed (FPS), flops, and cost are shown in the table below. The inference speed of IPSeg ($27.3$ FPS) is slightly lower than SSUL-M ($33.7$ FPS) and similar to CoinSeg-M ($28.2$ FPS). Due to the proposed image posterior branch, the model's floating-point operations ($137.1$ GFLOPs) are higher than the baseline ($94.9$ GFLOPs), and with an approximately $1$ GB increase in GPU usage.

| Method         | FPS      | FLOPs | GPU usage |
| :-------- | :--------: | :--:    | :--:        |
| IPSeg    |     27.3 |137.1G |       6.2G|
| SSUL-M   |     33.7 |94.9G  |       5.3G|
| CoinSeg-M|     28.2 |96.3G  | 5.6G      |

Overall, IPSeg introduces an additional image posterior branch with slight increases in model parameters, training and inference costs but brings great performance improvement. It is a worthwhile trade-off between performance and cost.

---

## 2. Manuscript Modification
To provide clearer insight into the revisions we make to our paper and the experiments conducted in response to the reviewers' feedback, we summarize the changes during the rebuttal period as follows：

### Additional analyses:
* We conduct a quantitative evaluation of image poserior branch on different settings. The results show that image posterior branch has excellent resilience against catastrophic forgetting. (reviewer zxHa Q1)
* We provide a comprehensive analysis of the model parameters, training, and inference costs of IPSeg compared with previous works. The results show that IPSeg introduces slight increases in model parameters, training and inference costs but brings great performance improvement. (reviewer zxHa Q4 and reviewer wQeR W2 and reviewer FU4w Q4)
* We conduct ablation study on different salient map. The results show that the default saliency map struggles with identifying "Stuff" classes, and high-quality salient map can obtain better performance. (reviewer zxHa Q5)

### Clarification:
* More details on the process to get the final prediction maps are provided in L269-L275. (reviewer wQeR Q2)
* More details on the construction of memory buffer are provided in the appendix. (reviewer FU4w Q2)
* Additional SOTA methods comparisons are added in Table 1 and Table 2. (reviewer wQeR W5)
* Table 2 is reorganized and the results of "IPSeg w/o M" are added for comprehensive and fair comparison. (reviewer wQeR W5, latest revision)

### Correction:
* A typo is fixed in Equation 3. ($\phi_{1:T}$ --> $\phi_{0:T}$)
* The statement in L156 is revised by adding previous work. (reviewer wQeR W1)
* All figures are replaced with PDF versions for better presentation. (reviewer FU4w Q3)

---

### Meta-Review · Area_Chair_BNMR · 2024-12-21

**Metareview:**

The paper addresses key issues in the task of class-incremental semantic segmentation, focusing on two main issues i.e., semantic drift and separate optimization that hinder performance. The proposed IPSeg method introduces two primary innovations: (1) Image Posterior Guidance, which leverages image-level classification to rectify segmentation errors and mitigate separate optimization, and (2) Permanent-Temporary Semantics Decoupling, aimed at distinguishing stable background and dynamic instance semantics. Extensive experiments on Pascal VOC and ADE20K datasets demonstrate that IPSeg achieves notable performance gains over SOTA methods, especially in long-term incremental learning scenarios.

Strengths:
The paper clearly identifies and addresses the issues of semantic drift and separate optimization, which are critical for improving CISS. IPSeg’s use of image posterior probabilities to guide segmentation is reasonable and is supported by extensive ablation studies and performance benchmarks. The experimental results consistently show that IPSeg outperforms baseline and competing methods across different datasets and settings. The authors' thorough responses to reviewer questions demonstrate the method’s robustness, particularly in scenarios with memory buffers. While the abstract may be somewhat confusing, the overall writing is generally clear in showing the main ideas and implementation.

Weaknesses:
IPSeg introduces several modules to address the observed issues, however, their entanglement makes it difficult to isolate the contributions of each component, limiting the clarity of the paper’s novelty and potential to inspire future work. The overall approach, though effective, is not entirely novel. Experimental results indicate that IPSeg performs well with a memory buffer; however, its performance without memory (IPSeg w/o M) is less competitive, lagging behind methods like LGKD+PLOP in data-free settings (noted by Reviewer wQeR). This discrepancy raises doubts about the core hypothesis that image-level classification mitigates forgetting better than segmentation. Additionally, the computational cost of IPSeg is higher than that of some baseline methods. Reviewers also highlighted the complexity of certain sections (e.g., semantic decoupling), suggesting that these areas could benefit from simplification and clearer comparisons with existing methods tackling similar challenges.

Decision:
The submission received mixed ratings, with two positive and two negative reviews. Reviewer k5js provided positive feedback but their comments were general, thus I slightly down-weighted of their evaluation in the final decision-making. Although the paper presents strong experimental results across multiple datasets, significant concerns remain regarding the novelty of IPSeg’s design and the empirical validation of its core components. As a result, the paper in its current form may not be accepted this time.

**Additional Comments On Reviewer Discussion:**

During the rebuttal period, the reviewers raised concerns primarily from the following three perspectives:
- (1) the performance of IPSeg without a memory buffer,
- (2) the computational cost compared to baseline methods, and
- (3) clarity in the explanation of the permanent-temporary semantics decoupling mechanism.

Reviewer zxHa questioned the effectiveness of image-level classification without memory buffers, suggesting that IPSeg's core hypothesis was not fully validated in such scenarios. In response, the authors provided ablation studies and new experimental results demonstrating that while IPSeg w/o M performs slightly below LGKD+PLOP, the primary contribution lies in the memory-buffer version, which achieves SOTA results. Reviewer wQeR expressed concerns about the computational cost, which the authors addressed by clarifying the trade-off between performance and inference time, where the increased complexity yields significant accuracy improvements. Additionally, FU4w requested clearer explanations of the decoupling strategy, which the authors addressed by revising relevant sections and providing updated visualizations. These responses were considered in the final decision, with the performance gap in memory-free scenarios remaining a key factor in the recommendation for rejection, despite the overall strength and innovation of the paper.

---

### Decision · Program_Chairs · 2025-01-22

Reject